# Thermogravimetric analysis of the co-combustion of coal and polyvinyl chloride

Hongbin Gao [1]◉, Jingkuan Li[2]◉*

1 Automation Department, Shanxi University, Taiyuan, PR China, 2 Power Engineering Department, Shanxi University, Taiyuan, PR China

◉ These authors contributed equally to this work.
* 13994259424@163.com

## Abstract

Coal gangue has the shortcomings of low calorific value and refractory burnout, while polyvinyl chloride has the advantages of a long combustion process and high calorific value. In order to make up for these shortcomings of coal gangue, the possibility of a treatment method based on co-combustion of coal gangue with polyvinyl chloride, which can be centrally recovered from municipal solid waste, is proposed. In order to analyze the combustion effect of a mixture of these two substances, experimental samples were prepared by mixing these two substances in three different ratios, and they were tested by thermogravimetric analysis. The experimental results were compared, analyzed and evaluated. The effects of the proportion of polyvinyl chloride in the mixture on the temperature parameters, activation energy, and interaction during co-combustion were analyzed. In order to analyze the interaction during co-combustion of the two, a coupling analysis method for mixed combustion is presented, and the effectiveness of this method is verified by comparing with the correlation analysis results of co-combustion. The results show that co-combustion can mitigate the ignition difficulty and burnout of coal gangue. When the proportion of polyvinyl chloride in the mixture was increased from 20% to 80%, the maximum weightlessness rate of the first stage rapidly increased from 4.5%/min to 15.6%/min; however, that of the second stage slowly increased from 3.7%/min to 4.2%/min. A 20% proportion of polyvinyl chloride showed the most significant promotion of co-combustion, with a maximum coupling coefficient of 0.00318, which was 1.11 and 1.35 times greater than that of 50% and 80% proportions, respectively. Co-combustion can reduce the activation energy of coal gangue during the initial and end stages. Therefore, co-combustion is helpful to improve the problems of low calorific value and refractory burnout of coal gangue.

**Data Availability Statement:** All relevant data are within the paper and its Supporting Information files.

**Funding:** This work is financially supported by the Excellent Talents Scientific and Technological

## Introduction

Coal gangue (CG) is a common industrial solid waste. An economical, reliable, and comprehensive method to utilize this waste is to burn it for electricity generation [1,2]. Owing to its ignition difficulty, unstable combustion, and low calorific value [3], researchers worldwide have investigated the combustion and co-combustion characteristics of CG. Recently, the co-

Innovation Foundation of Shanxi Province (No. 201805D211039).

**Competing interests:** The authors have declared that no competing interests exist.

**Abbreviations:** ad, air dried; CG, coal gangue; DSC, differential scanning calorimetry; DTG, differential thermogravimetry, FC, fixed carbon; LHV, lower heating value; PVC, polyvinyl chloride; TG, thermogravimetric; TGA, thermogravimetric analysis.

combustion characteristics of CG was discussed in terms of the various properties of coal rather than non-coal substances [4–7]. Several scholars have conducted research on the co-combustion characteristics of CG with coal gas [8–12]; their results showed that the mixing ratio is the most important parameter to consider when optimizing the efficiency of co-combustion. Based on these research results, different proportions of mixed samples are prepared for the experiment, and the characteristic parameters, activation energy, and coupling degree of co-combustion are analyzed according to the experimental results in this paper. Ren *et al.* obtained different conclusions about the influence of coal gas on the combustion of CG in a static bed [13]; they showed that the gas could inhibit or slow down the combustion reaction of gangue with oxygen and cause insufficient combustion. This can be used as a reference for the analysis of co-combustion characteristics at different temperatures. The research on the co-combustion of low-rank coal with woody biomass and miscanthus by Metovic *et al.* suggested that co-firing Bosnian coals with woody sawdust and Miscanthus can enable higher co-firing ratios for pulverized combustion by Kazagic, A. [14]. Wang analyzed the co-combustion of CG, which had a fixed carbon content of 33.35%, with sludge by using thermogravimetric analysis and categorized the weight loss process involved in the burning of CG into three stages [15]. In this study, the combustion weightlessness process of CG adopts such a piece-wise method. Li *et al.* found that the combustion of gangue mixed with NaCl promoted the precipitation of HCl significantly, with the average emission concentration of HCl in the flue gas reaching 56.32 mg/m$^3$, which exceeds the upper limit of the emission standard [16]. The samples used in this experiment contain halogen elements; therefore, the flue gas produced by the experiment requires special treatment. Meng *et al.* found that the pyrolysis of CG in a $CO_2$ environment can be divided into three stages: moisture release, devolatilization, and char gasification by $CO_2$ in a higher temperature zone [17]. Zhang showed that both the burnout temperature and peak temperature of the pyrolysis coke combustion stage in the combustion of gangue mixed with pine sawdust decreased with an increase in the proportion of pine sawdust [18]. However, the reasons for the reduction were not analyzed in detail. Kan *et al.* investigated the combustion of CG mixed with rubber sawdust and found that the co-combustion of the two was helpful to the burnout of CG, but more PM2.5 was produced in the flue gas. It can be seen that the addition of other substances to the combustion of CG may promote or inhibit the combustion effect, and the effect may be closely related to the mixing ratio. In order to overcome the shortcomings of low calorific value and refractory burnout of CG, the mixed combustion material needs to have the characteristics of high calorific value and long combustion process. [19].

Plastic is a type of common municipal solid waste. If the plastic present in municipal solid waste (which is exposed to the environment) is burned, it produces toxic, harmful, and malodorous gases that enter the atmosphere, which results in serious damage to public health [20]. During the renovation of old cities as well as during the renewal of urban pipe networks, large amounts of polyvinyl chloride (PVC) plastic waste pipes are often concentrated as municipal solid waste [21]. Such a high concentration of PVC makes it possible to perform renewal as a single process. The pyrolysis, co-pyrolysis, and combustion properties of various types of municipal solid waste and common plastics have been researched by scholars all over the world [22–35]. Their studies have shown that most plastics exhibit advantages such as high volatility, low ash content, and high calorific value. The combustion of CG mixed with PVC may make up for the shortcomings of low calorific value and refractory burnout of CG and may also improve the poor economy caused by the small incineration scale of combustible solid waste. Therefore, the co-combustion characteristics of the two are worth studying.

In this study, the TG experimental curves of three proportions of CG and PVC blends are evaluated and compared with those of CG and PVC. The influences of the PVC ratio in the

mixture on the temperature parameters, activation energy, and interaction of co-combustion are analyzed. In order to effectively analyze the interaction between CG and PVC in the overall co-combustion process, an analysis method for the coupling of the two combustion processes, which is based on the fact that the variance between the actual value and the mean value can be calculated by the least-squares method, is presented. The analysis results are obtained, and the co-combustion effect of CG and PVC is evaluated according to the analysis results.

## Experimental

### Selection and preparation of experimental samples

The CG used in this experiment was obtained from a CG power generation company in Shanxi Province, China. Different types and proportions of plasticizers, colorants, heat stabilizers, and other additives are often added during the production of plastic products to modify their physical and chemical properties. To generalize this experimental study, the basic raw PVC material produced by a petrochemical company without any additives was selected as the sample source.

A grinding machine was used to grind the raw materials of CG and PVC; then, the particles that could pass through #200 were taken as the experimental samples. The prepared samples were placed in a vacuum drying box and dried at 80˚C for 10 h and then sealed in a bag for preservation. The results of the proximate and ultimate analyses of the prepared PVC and CG samples are shown in Table 1.

In order to improve the representativeness of the samples, it was ensured the quality of the prepared samples was at least three times the quality of the required samples, and the mixing, agitating, and grinding time was not less than 5 min. After the samples were fully mixed, approximately 20 mg of it was taken for the co-combustion experiment.

### Selection of experimental equipment and system design

Thermogravimetric analysis (TGA) is a type of thermal analysis technology that uses a thermobalance to estimate the relationship between the mass and temperature under a programmed control temperature. In this study, a TG-DTG/DSC thermogravimetric analyzer with a differential thermal compensation function (SETARAM, France) was used in the experiment on combustion characteristics. The sensitivity of the balance used in the analyzer is 0.1 μg. Because the flue gas from the combustion of PVC contains a higher concentration of HCl gas, which causes corrosion in the air and overflow equipment, the experiments on the emission of HCl and dioxins from the co-combustion of CG with PVC were carried out in a self-made horizontal fixed bed test system. (The study on the flue gas emission law will be not described in this paper; it will be described in other subsequent articles.)

The schematic diagram and photo of the experimental system are shown in Fig 1(A) and 1 (B), respectively. The system is composed of a pipe furnace, gas supply equipment, and tail gas

**Table 1. Results of proximate and ultimate analyses of the prepared polyvinyl chloride (PVC) and coal gangue (CG) samples.**

| Sample | Proximate analysis ad (wt%) | | | | Ultimate analysis ad (wt%) | | | | | LHV |
|--------|------|------|-------|------|------|------|--------|------|------|---------|
| | M | V | FC | A | C | H | O[a] | N | S | (MJ/kg) |
| CG | 1.04 | 15.4 | 22.4 | 61.2 | 26.1 | 1.42 | 7.39 | 0.43 | 1.42 | 9.372 |
| PVC | 1.86 | 76.4 | 19.02 | 2.74 | 31.7 | 7.31 | 55.7[b] | 0.31 | 0.41 | 23.526 |

ad–air dried basis; LHV–lower heating value; M–moisture; V–Volatiles; FC–Fixed Carbon; A–Ash

[a]mean subtraction

[b]mean chlorinity

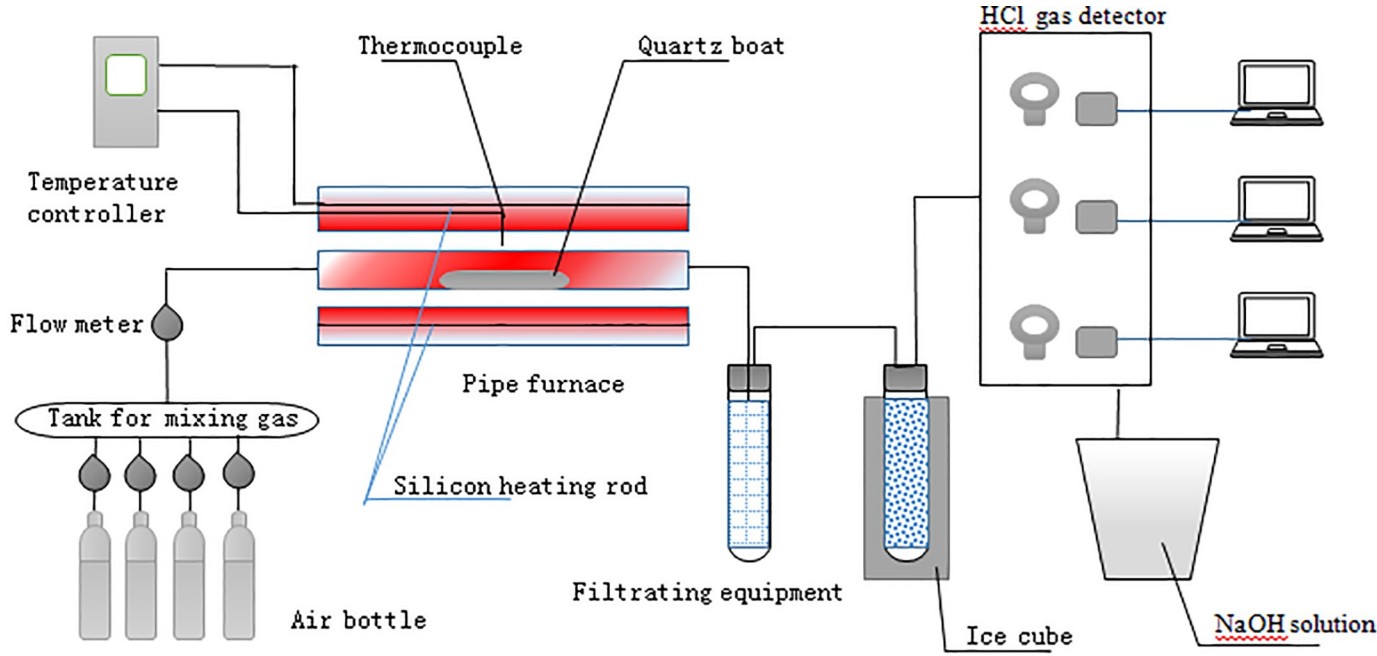

(a) Experimental schematic diagram

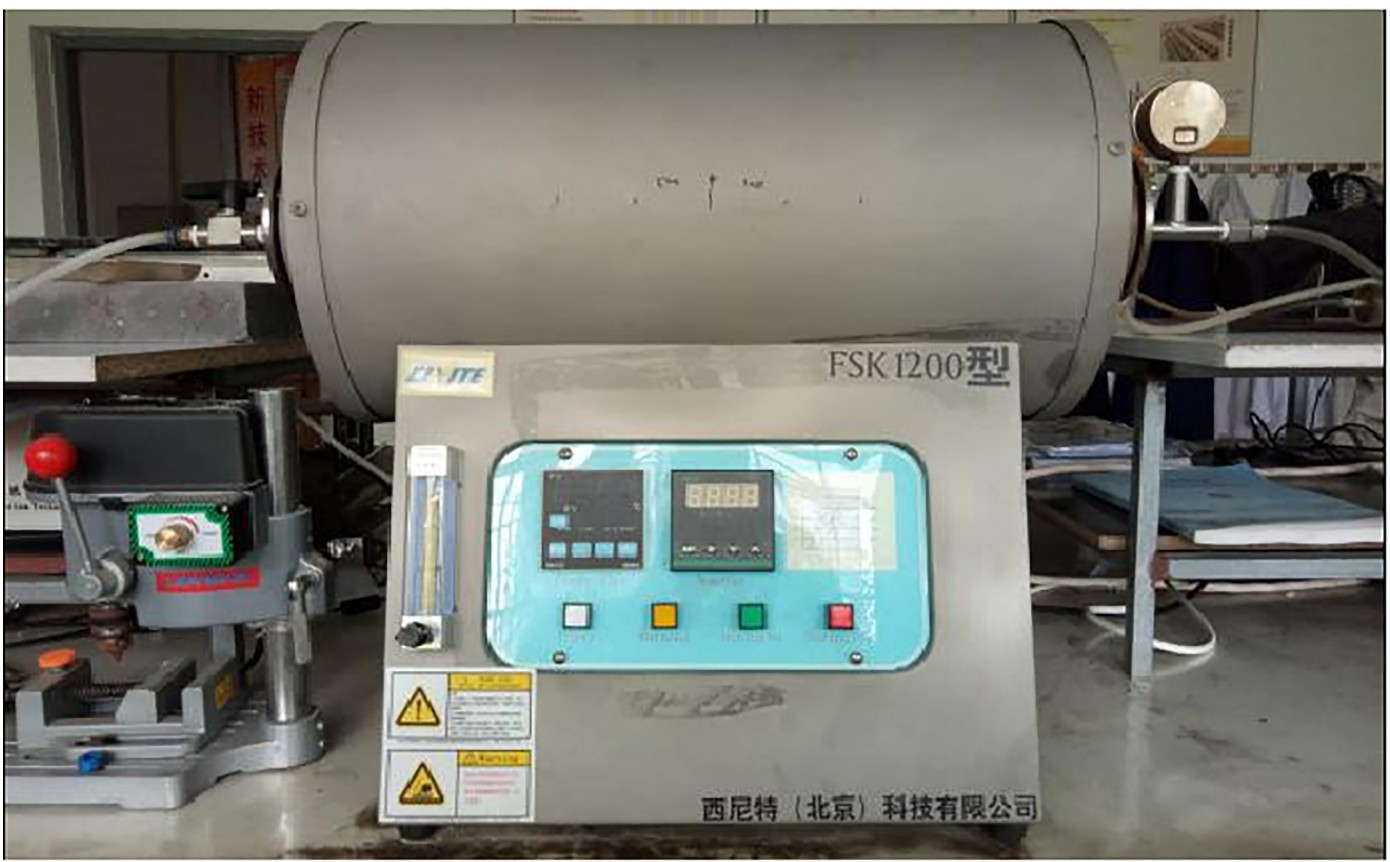

(b) Photo of experimental equipment

**Fig 1. Experimental system diagram.**

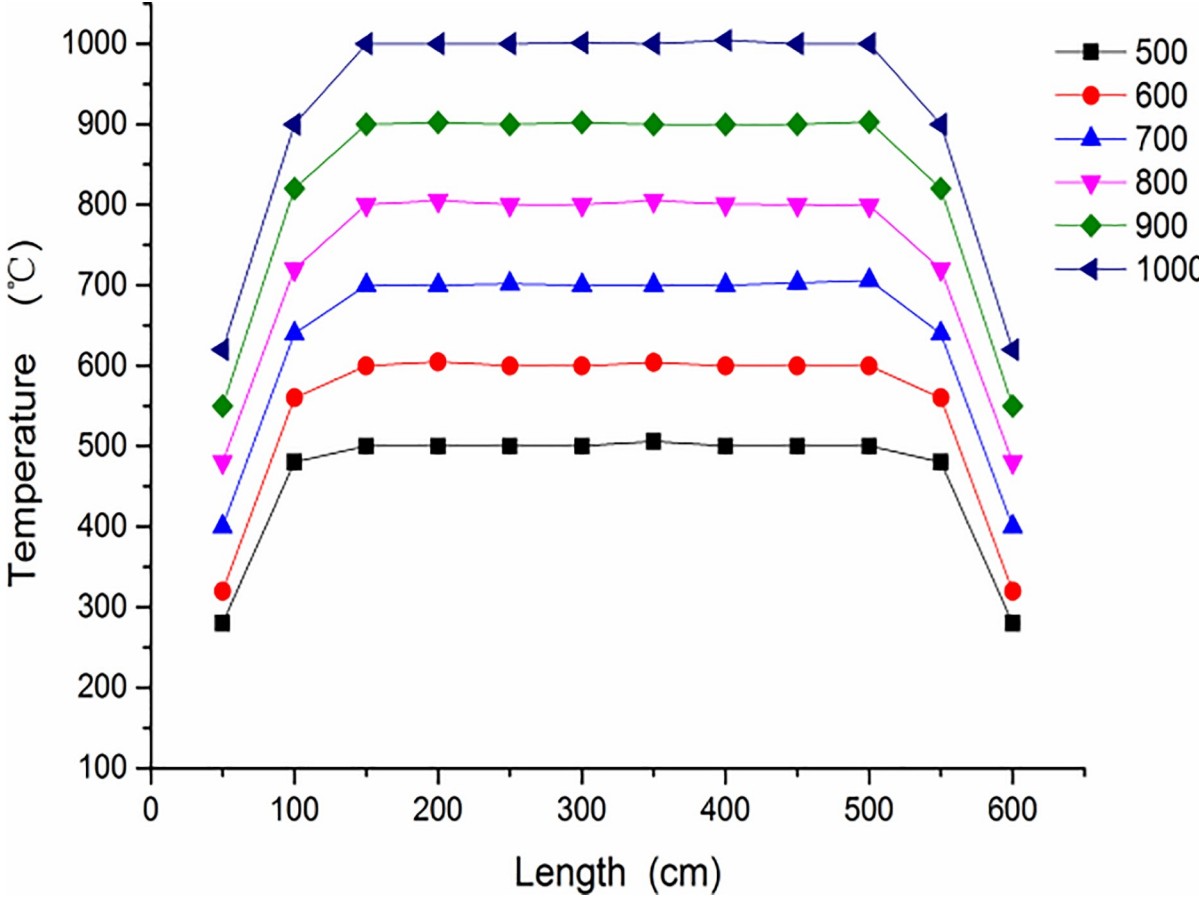

**Fig 2. Calibration results of experimental system.**

detection and treatment equipment. The heat generated by the silicon heating rod is adjusted by the temperature controller in the tube furnace to meet the temperature requirements in the experimental scheme. The reactor is a quartz tube with a diameter of 50 mm and a length of 650 mm. After weighing, the experimental samples were placed in the quartz boat and put together in the middle of the reactor, and the two sides were connected to the inlet and outlet gas through flanges to ensure the sealing of the system.

In order to improve the reliability of the experimental results, the constant temperature section of the quartz tube reactor in the tubular furnace was calibrated before the experiment. The calibration results are shown in Fig 2. The temperature of the 200 mm long section in the quartz tube reactor is kept constant in the range of 500–1000°C. In the experiment, the gas in the cylinder is discharged into the atmosphere after innocuous treatment. The dioxins produced in the experiment will be treated specially.

## Experimental scheme

**Single-component combustion experiment.**   In the experiment, approximately 10 mg of the sample was placed in the crucible of the thermogravimetric analyzer. The gas flow rate was 60 mL/min. Therefore, the main stage of the reaction can be regarded as a pure chemical kinetic reaction. At the beginning of the experiment, the temperature was increased to 50°C for 10 min in order to reduce the effect of external disturbances on the experimental data.

**Table 2. Experimental scheme for single-component combustion.**

| Experiment | Sample | Heating rate (˚C/min) | Atmosphere | Gas flow rate (mL/min) | Heating range (˚C) |
|---|---|---|---|---|---|
| Combustion | CG, PVC | 10 | Air | 60 | 20–1000 |
| Pyrolysis | CG, PVC | 10 | $N_2$ | 60 | 20–1000 |
| Combustion dynamics | CG, PVC | 5<br>10<br>30 | Air | 60 | 20–1000 |

CG is coal gangue and PVC is polyvinyl chloride

Then, the temperature was increased from 50˚C to 1000˚C (because the burnout temperature ranges of CG and PVC are 550˚C– 600˚C), and maintained at this value for 1 h, and lowered to approximately 20˚C (room temperature). The experimental scheme is shown in Table 2.

**Mixed-component combustion experiment.** In the mixed-component combustion experiment, the variation of the co-combustion characteristic parameters and the coupling mechanism involved in the combustion of CG mixed with PVC were investigated. The experimental scheme is shown in Table 3. The experiments on the co-combustion law were carried out at different mixing proportions and heating rates. The procedure of the mixed-component combustion experiment was the same as that of the single-component combustion experiment.

In these experiments, about 18 mg oxygen is needed for 10 mg gangue combustion and about 32 mg oxygen is needed for 10 mg PVC combustion. Taking the heating rate of 30˚C/min and air flow rate of 60 mL/min as an example, when the temperature rises from 50˚C to 1000˚C, more than 300 mg oxygen needs to be provided to the system. It can be seen that the experimental gas supply is much larger than the amount of air needed for combustion.

## Analysis method

**Combustion characteristic parameter analysis.** The characteristic parameters used in this study to characterize the co-combustion process include the temperature corresponding to the maximum rate of weightlessness ($T_p$), ignition temperature ($T_i$), burnout temperature ($T_f$), maximum weightlessness value ($TG_{max}$), and maximum weightlessness rate ($DTG_{max}$).

The above five parameters for the combustion of CG, combustion of PVC, and co-combustion with three different proportions can be obtained using the TG/DTG-$T$ curve from the experiments. By comparing the five characteristic parameters of combustion of only CG or PVC with those of co-combustion of CG and PVC in three different proportions, the change law of co-combustion parameters can be obtained.

**Table 3. Experimental scheme of co-combustion experiment.**

| Experiment | Sample | Atmosphere | Heating rate (˚C/min) |
|---|---|---|---|
| Different mixed proportions | CG20PVC80<br>CG50PVC50<br>CG80PVC20 | Air | 10 |
| Different heating rates | CG50PVC50 | Air | 5<br>10<br>30 |

CG is coal gangue and PVC is polyvinyl chloride

**Combustion dynamics analysis.** To reveal the inherent relationship in the chemical reaction between CG and PVC in co-combustion, the activation energies in the individual combustion cases and co-combustion cases were obtained from the measured experimental data. The distributed activation energy model can be used to analyze the complex co-combustion reactions caused by the inhomogeneity of combustion [36,37].

The distributed activation energy model can be expressed as shown in Eq 1:

$$1 - \frac{w}{w_0} = \int_0^\infty \exp(-A\int_0^t \exp(-E/RT)dt)f(E)dE, \tag{1}$$

where $w$ is the mass of solid reactants varying with time, $w_0$ is the mass of solid reactants at the end of the reaction, $E$ is the activation energy, $R$ is the universal gas constant, $f(E)$ is the distribution function for the difference in activation energy of the first-order irreversible reaction, $A$ is the frequency factor corresponding to the activation energy $E$, $T$ is the reaction temperature, and $t$ is the reaction time.

According to the Miura integral method [38], Eq 1 can be simplified as Eq 2:

$$\alpha = \frac{w}{w_0} = 1 - \int_0^\infty \Phi(E, T)f(E)dE, \tag{2}$$

where $\alpha$ is the conversion ratio, and $\Phi(E,T)$ is the relation between the activation energy and temperature (implicit) function. Eq 3 can be obtained from Eq 1 and Eq 2:

$$\Phi(E, T) = \exp(-A\int_0^t \exp(-E/RT)dt). \tag{3}$$

To establish the relationship between time $t$ and reaction temperature $T$, $\Phi(E,T)$ can be approximated by the step function $E = E_S$ according to the Miura integral. Thus, Eq 4 and Eq 5 can be obtained as follows:

$$\Phi(E, T) \cong \exp\left(-\frac{A}{\beta}\int_0^T \exp(-E/RT)dT\right) \cong \exp\left(-\frac{ART^2}{\beta E}\exp(-E/RT)\right) \tag{4}$$

and

$$\frac{w}{w_0} \cong 1 - \int_{E_s}^\infty f(E)dE = \int_0^{E_s} f(E)dE \tag{5}$$

where $E_S$ is the critical activation energy at which the substance can react.

Eq 5 approximates the reaction by including only one $Es$, and its mathematical expression is as follows:

$$dw/dt \cong d(\Delta w)/dt = A\exp(-E/RT)(\Delta w_0 - \Delta w), \tag{6}$$

where $\Delta w$ is the weightlessness mass of the solid reactants varying with time, and $\Delta w_0$ is the weightlessness mass of the solid reactants at the end of the reaction.

The total rate can be approximated by the rate of the $j$-th reaction, and Eq 6 can be expressed as

$$1 - \Delta w/\Delta w_0 = \exp(-A\int_0^t \exp(-E/RT)dt) \cong \exp\left(-\frac{ART^2}{\beta E}\exp(-E/RT)\right). \tag{7}$$

Eq 8 can be derived from the mathematical transformation of Eq 7:

$$\ln(\frac{\beta}{T^2}) = \ln(\frac{AR}{E}) - \ln\left\{-\ln\left(1 - \frac{\Delta w}{\Delta w_0}\right)\right\} - \frac{E}{R}\frac{1}{T}. \tag{8}$$

Taking $1 - \Delta w/\Delta w_0 = \Phi(E,T) \approx 0.58$ for all reactions [37], the simplified model can be described as follows:

$$\ln(\frac{\beta}{T^2}) = \ln(\frac{AR}{E}) + 0.6075 - \frac{E}{R}\frac{1}{T}. \tag{9}$$

Both $E$ and $A$ can be obtained from the Arrhenius curves of $\ln(\beta/T^2)$ and $1/T$ under the same conversion ratio. The activation energy at some conversion ratio can be obtained from the slope of the straight line. The value of the frequency factor corresponding to the activation energy can be obtained from the straight-line intercept.

**Mixed-combustion coupling analysis method.** The characteristics of the mixed combustible are not simply a superposition of the individual combustion characteristics of each component. The structure and combustion characteristic parameters of each component in the mixture are different. There are also differences in the endothermic and exothermic stages from heating to burnout. Furthermore, there is an interaction between the components in the combustion process. These factors may promote or suppress the mixing process [39,40].

In this study, the linear superposition method was used to analyze the interactions among the components. First, the TG curves of single-substance combustion were obtained under the same experimental conditions. If the combustion of CG and PVC in the mixture were independent, that is, if there was no interaction between CG and PVC in the mixed-combustion process, the TG curve obtained from the mixed combustible would be the linear superposition of the TG curves from each component [41,42]. The theoretical TG curve of the mixed combustibles can be expressed according to Eq 10.

$$TG_{cal}(T) = x_1 * TG_A(T) + x_2 * TG_B(T), \tag{10}$$

where $TG_{cal}(T)$ is the calculation value of the theoretical weightlessness mass for mixed combustibles, $TG_A(T)$ is the experimental weightlessness mass of combustible A, $TG_B(T)$ is the experimental weightlessness mass of combustible B, $x_1$ is the mass fraction of combustible A in the mixed combustible, and $x_2$ is the mass fraction of combustible B in the mixed combustible, where $x_1 + x_2 = 1$.

If there is no interaction between combustible A and combustible B,

$$TG_{cal}(T) = TG_{exp}, \tag{11}$$

where $TG_{exp}(T)$ is the experimental weightlessness mass of the mixed combustible.

If there is a new product in the co-combustion,

$$TG_{cal}(T) \neq TG_{exp}. \tag{12}$$

This indicates that combustible A and combustible B interact with each other in the combustion process. When comparing data, the error between the calculated data and actual data is usually solved using the least-squares method. Eq 13 represents the residual sum of squares function S(T):

$$S(T) = [TGcal_{exp}]^2. \tag{13}$$

To quantify and normalize the magnitude of interaction between the two combustibles in mixed combustion, the experimental and theoretical weightlessness masses were used to replace the corresponding residual, and the residual was nondimensionalized to obtain the temperature-change co-firing thermal coupling coefficient as follows:

$$\lambda(T) = \frac{[TG_{\exp}(T) - x \cdot TG_A(T) - (1-x) \cdot TG_B(T)]^2}{TG_{\exp}(T) \cdot [x \cdot TG_A(T) + (1-x) \cdot TG_B(T)]}. \tag{14}$$

The increasing $\lambda$ value indicates that the interaction between the two combustibles is enhanced. Conversely, a decrease in the value indicates a weakening of the coupling between the two combustibles.

By averaging the value of $\lambda$ according to temperature (time), the temperature-averaged co-combustion thermal coupling coefficient $\bar{\lambda}$ can be obtained as follows:

$$\bar{\lambda} = \frac{1}{T_h - T_s} \int_{T_s}^{T_h} \frac{[TG_{\exp}(T) - x \cdot TG_A(T) - (1-x) \cdot TG_B(T)]^2}{TG_{\exp}(T) \cdot [x \cdot TG_A(T) + (1-x) \cdot TG_B(T)]} dT \tag{15}$$

where $T_s$ is the corresponding temperature value of the theoretical and experimental TG curves at the first separation, and $T_h$ is the corresponding temperature value of the theoretical and experimental TG curves at the final merger.

The properties of interaction can be determined from the coupling directional discriminant ($\Pi$), which is calculated as follows:

$$\Pi = \int_{T_s}^{T_h} \int [TGA_{B_{exp}}[]]. \tag{16}$$

If $\Pi > 0$, the overall performance is viewed as a promotion. Conversely, $\Pi < 0$ represents the overall performance as an inhibition.

## Results and discussion

### Experimental results and discussion on separate combustion of CG and PVC

**Combustion process analysis.** The thermogravimetric (TG) curves of the thermogravimetric experimental results of CG and PVC are shown in Fig 3. The TG curves of CG and PVC can be roughly divided into three stages under an air or nitrogen atmosphere. The results of the three stages of weightlessness mass of CG are consistent with those from the aforementioned studies [4–7,15]; therefore, only the experimental results of PVC combustion and pyrolysis are analyzed.

The first phase is the quality maintenance phase. This is related to the fact that the chemical bonds in PVC are mainly C–C and C–H single bonds, while there are fewer unsaturated double bonds. Avila *et al.* explained the reactivity of fuel according to the mass gain and categorized the process into three types: the stable carbon-oxygen compound, temporary-type carbon-oxygen compound, and non-reactive carbon-oxygen complex [43]. Among them, the stable carbon-oxygen compound increases the activation energy of combustion, resulting in an increase in ignition temperature. Temporary-type carbon-oxygen compounds are highly susceptible to desorption, which reduces the activation energy of volatiles and increases the tendency of combustible volatiles.

The second stage is rapid weightlessness. With the increase in temperature, there will be significant weightlessness. Pyrolysis is considered to cause continuous cracking of the side chain, bridge bonds, and functional groups around the basic structural units and other thermally

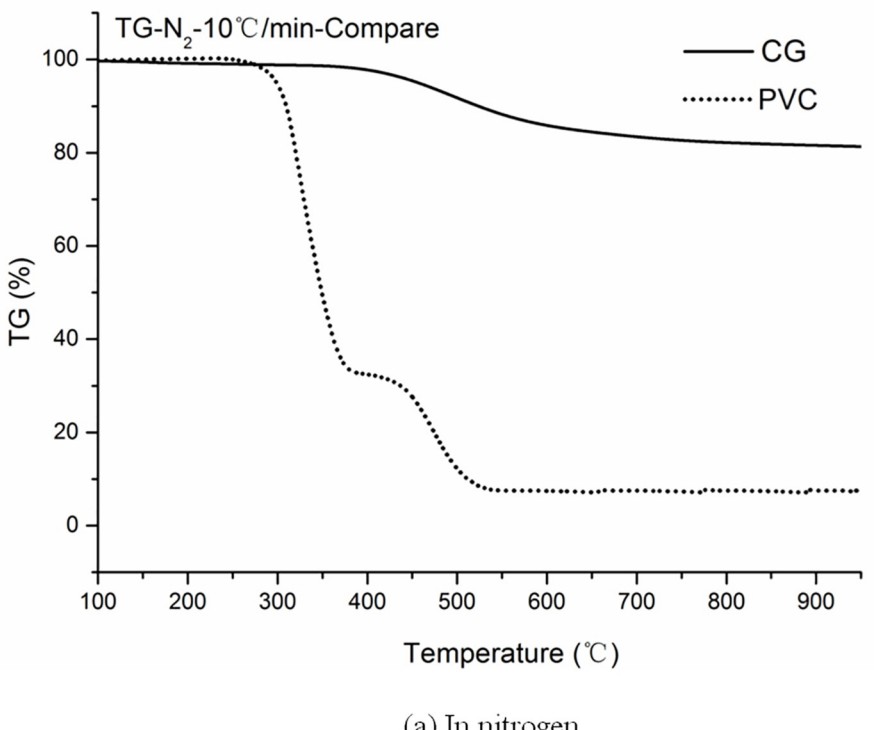

(a) In nitrogen

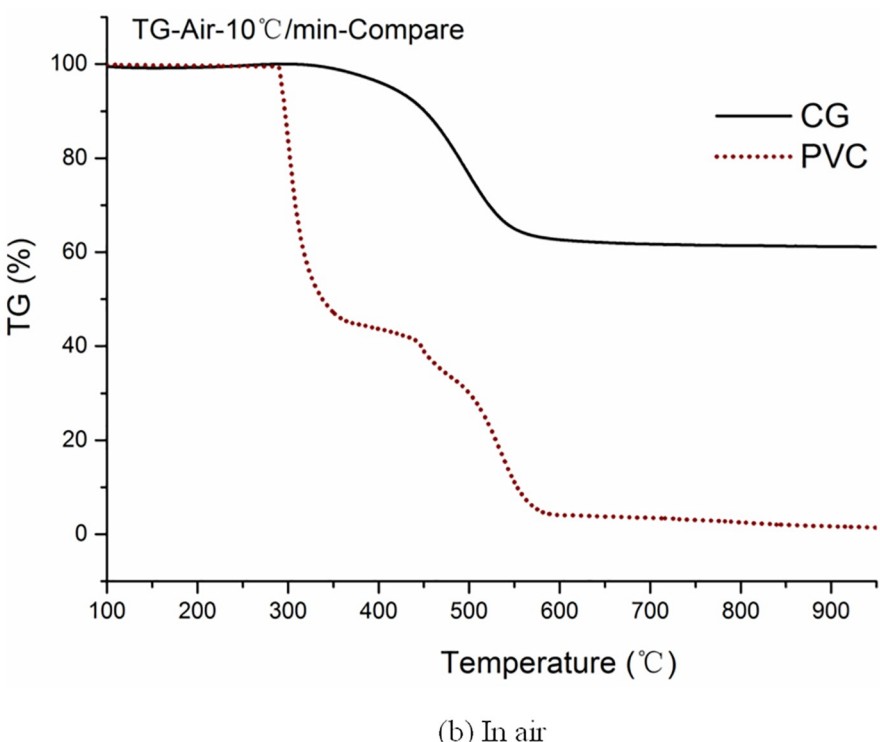

(b) In air

**Fig 3.** Thermogravimetry curves of combustion and pyrolysis for coal gangue (CG) and polyvinyl chloride (PVC) in (a) nitrogen and (b) air.

unstable components, forming low-molecular compounds that can escape. Solid-state products (semi-coke or coke) are formed by the condensation of basic structural units and the condensation of aromatic nuclei, which have good thermal stability. The process of combustion is more violent than pyrolysis, with overlapping volatilization analysis, volatile combustion, and coke combustion. The functional groups or side chains of the macromolecules in the PVC components cross the activation energy barrier and generate small molecular hydrocarbon gases. The volatility of combustibles is much higher than that of gangue; hence, the TG diagram shows rapid weight loss at the initial stage of combustion.

The third stage is the stage of slightly decreasing or constant weight. In this stage, on the one hand, a more stable ring structure is formed because of the condensation of PVC during pyrolysis or combustion, while on the other hand, carbon black with a graphite-like structure is produced because of local hypoxia or a local drop in temperature caused by airflow in the combustion stage. The ring and carbon black structures are relatively dense, and the heat and mass transfer occur slowly; therefore, burn out is difficult. In addition, because the ash content of PVC is very small, the volume in the combustion process decreases continuously, and a sharp decrease in the specific surface area and specific pore volume can lead to burnout difficulties.

Compared with the solid residue of combustion, the weight loss of pyrolysis is usually lower than that of combustion because of the formation of coke or semi-coke after pyrolysis. Both CG and PVC reflected this characteristic.

**Thermal analysis of reaction formation.**   By using the differential thermal compensation function of the TG-DTG/DSC thermal analyzer, the DSC curve of each sample can be drawn according to the experimental results. The convexity and depression of the curve represent the endothermic and exothermic conditions, respectively, the peak area represents the total amount of endothermicity and exothermicity, and the sharpness of the peak represents the endothermic and exothermic rate in the reaction process.

Fig 4 shows the DSC curve of CG and PVC combustion under air atmosphere. In the initial stage of the reaction, the volatiles were separated out. On the one hand, the pyrolysis reaction of separating out volatiles is a process that absorbs heat; on the other hand, the volatiles and oxygen oxidize slowly and release heat [44]. The comprehensive effect of the two is shown in the figure as the change in the DSC curve. With the increase in temperature, the formation rate of volatiles decreases gradually, and the reaction rate of oxidation increases. In the diagram, the DSC curve shows a gradual increase. In the later stages of combustion, the combustible components show a decreasing trend, and the contact area between oxygen and the combustible components also decrease, resulting in a decrease in the DSC curve after the formation of a peak. Because of the difference in chemical structure and combustion process, the two types of combustibles exhibit different peak shapes and numbers.

Clear endothermic peaks can be observed in the DSC curve of PVC, indicating that the heat required in the initial stage of its pyrolysis was much higher than that of CG. On the one hand, a considerable amount of energy is required for chemical bond fracture because of the high rate of weightlessness; on the other hand, Cl is basically separated out in the form of HCl before 400°C [45]. As the Cl content is 55.7%, the ratio of the combustible matrix is small, and the oxidation rate is low during pyrolysis. When the temperature rose to approximately 290°C, the pyrolysis peak was reached, and the endothermic energy reached its extreme value. With the increase in temperature and the amount of released gas, the combustible gas that was separated out caught fire and began to release heat, which increased to the maximum as the combustion rate increased. As the burning rate increases, the amount of released heat increases to the extreme value. With the reduction in the gases that were separated out, the release of heat began to slow down. At this time, the temperature reached the ignition temperature of coke in

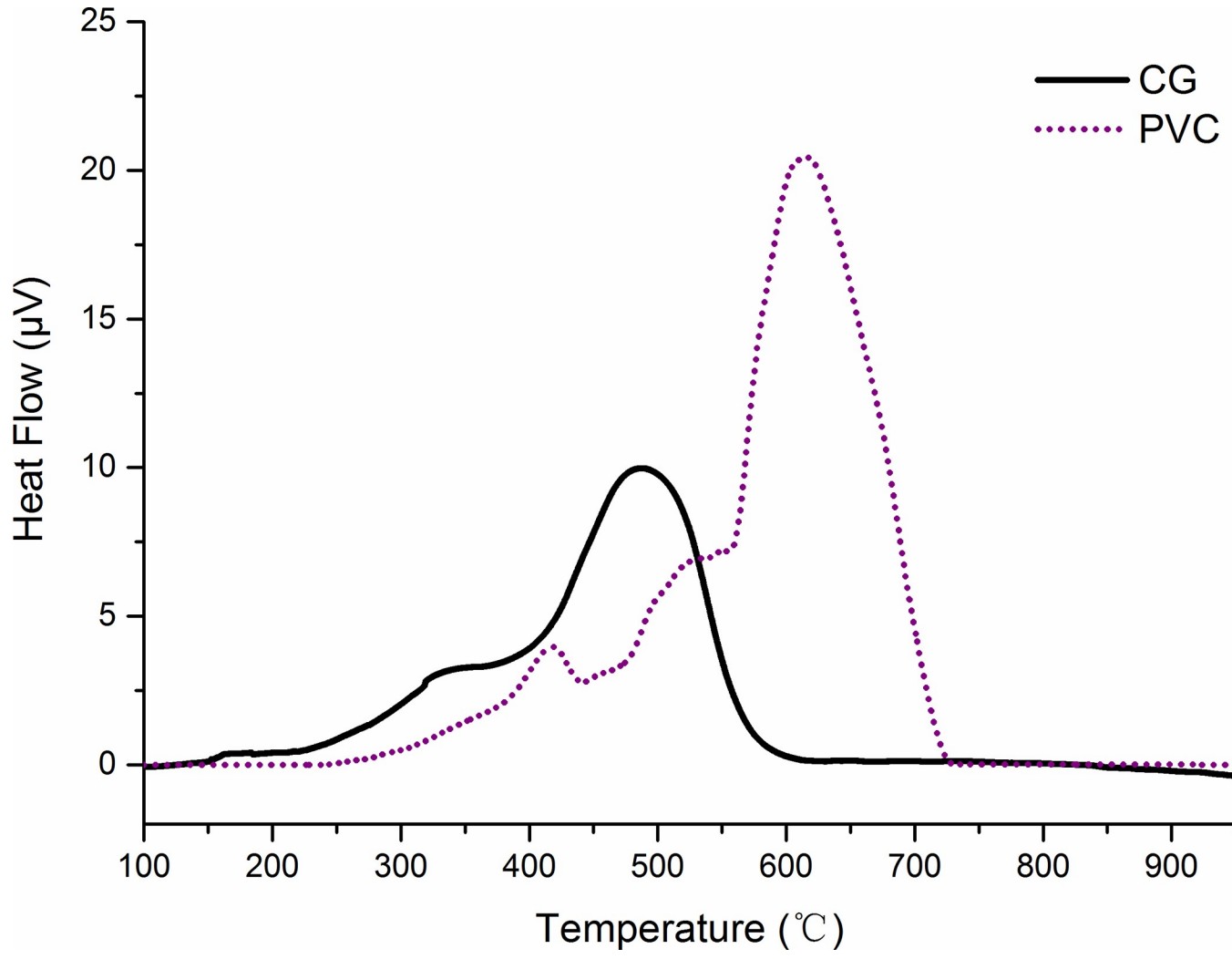

**Fig 4. Differential scanning calorimetry curves of flues in air atmosphere; CG and PVC are coal gangue and polyvinyl chloride, respectively.**

the pyrolysis organism; thus, the coke began to burn and release heat. Another exothermic peak appeared at the intersection of the reduced exothermic emission of separated gases and increased exothermic heat of coke. The peak value of exothermic heat appeared at about 450˚C. Then, the exothermic heat of combustion of the separated gases decreased rapidly, and the exothermic heat of coke combustion increased to the maximum value, at a temperature of approximately 535˚C.

Compared with PVC, CG released more total heat before ignition, which was conducive to its preheating. The long duration and relative smoothness of the heat release caused the combustion process to be more stable, reducing the thermal shock to the equipment, and was also beneficial to the adjustment and stabilization of the combustion process under low load conditions. However, the long combustion time, difficulty in burnout, and slow reaction rate make it difficult to achieve rapid adjustment. This also makes it difficult to adapt to the demand of rapid load changes.

**Analysis of the influence of heating rate.** The thermogravimetric experiments were carried out at heating rates of 5˚C/min, 10˚C/min, and 30˚C/min, and the results are shown in Fig 5. It can be seen that the weightlessness curve and the weightlessness rate curve shift

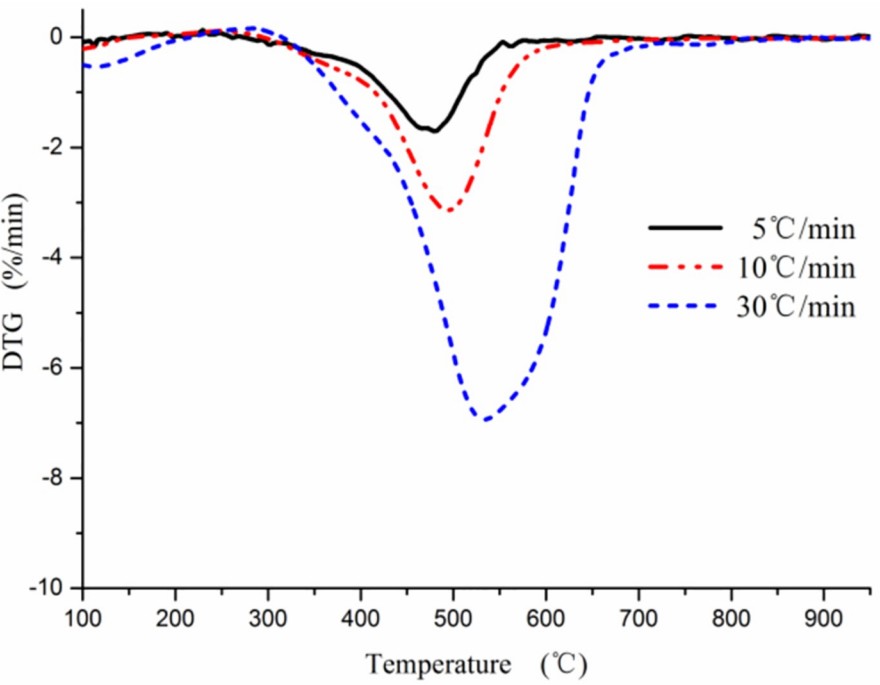

(a) Differential thermogravimetric curves of coal gangue

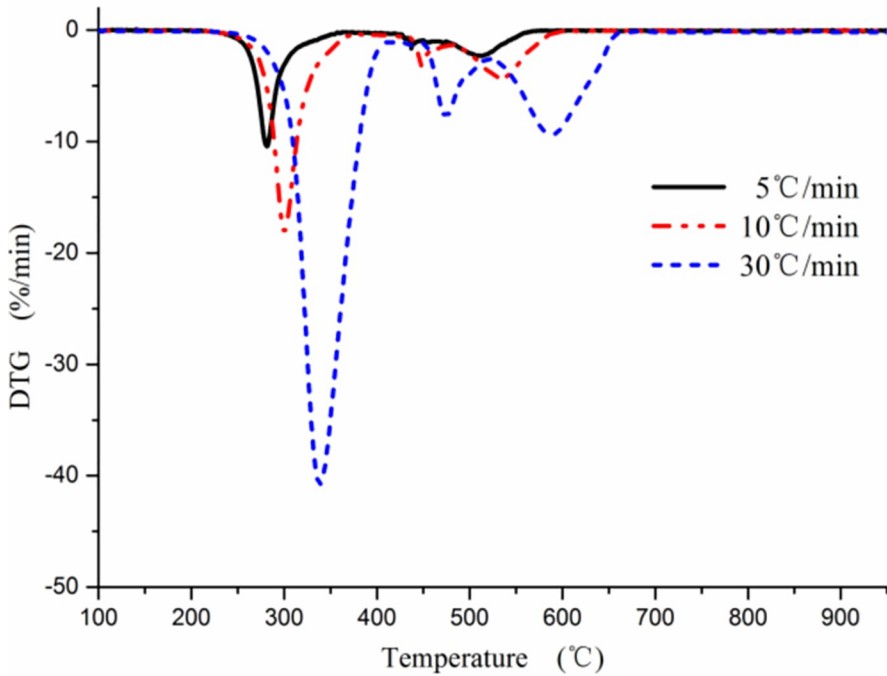

(b) Differential thermogravimetric curves of polyvinyl chloride

**Fig 5. Effect of heating rate on single group combustion.**

toward the high-temperature region after the heating rate is increased, and the reasons are as follows.

One reason is the reaction temperature. The higher the heating rate, the higher the reaction temperature, the higher the reaction rate, and the lower the partial pressure of oxygen at the gas-solid reaction interfaces, which forms a new reaction equilibrium. The weightlessness curve and weightlessness rate curve shift to the high-temperature region.

Another reason is the heating process of solid particles. First, the surface of the particles is heated by ambient heat radiation; next, the interior of the particle is heated through the process of heat conduction. When the reaction environment is in a warming process, there is always a temperature difference between the measured temperature and the average temperature of the reactants. With an increase in the heating rate, the temperature gradient inside the reactor increases on the surface of the particle and inside the particle. These behaviors lead to an increase in the gap between the measured temperature and the actual reaction temperature. The weightlessness curve and the weightlessness rate curve shift to the high-temperature region.

With the increase in heating rate, the maximum weight loss rate of PVC increased significantly, from 10.2%/min at 5˚C/min to 18.1%/min at 10˚C/min and 41.3% at 30˚C/min, increasing 1.8 times and 4 times, respectively.

**Combustion characteristic parameter analysis.** The combustion characteristic parameters of the two substances can be determined according to the TG-DTG diagram curve, as shown in Table 4. It can be seen that one or more combustion weightlessness peaks appear in the combustion process because of the different properties of the combustibles. CG exhibited only one weightlessness peak, but PVC exhibited two weightlessness peaks, which can be described in two stages using the DSC curve. There was no clear exothermic peak on the DSC curve corresponding to the weightlessness peak in the first stage, indicating that the pyrolysis reaction took place mainly in this stage. In the second stage, the DSC curves corresponding to the weightlessness peaks had a clear exothermic peak, which suggests that there was a severe combustion reaction.

Table 4 shows that the temperature corresponding to the first weightlessness peak of PVC is much lower than that of CG, and the temperature is positively correlated with the volatile content of the combustible. The higher the volatile content is, the more likely it is to react with oxygen when it is released. On the one hand, the C/H ratio of CG is 0.6 and that of PVC is 1.9. Thus, the difference in the C/H ratio between the two is rather large. On the other hand, it is related to the binding form of hydrogen in the combustible. The hydrogen structures in CG include hydroxyl groups, aromatic hydrocarbons, and hydrogenated aromatic hydrocarbons, whose bindings are more stable, while the hydrogen structures in PVC mainly include methyl ($-CH_3$), methylene ($-CH_2-$), alkyl chains ($-CH-$), and other forms, in which the chain cracking occurs easily at lower temperatures, forming a volatile component.

**Table 4. Combustion characteristic parameters of single combustible.**

| Sample | $T_i$ (˚C) | $T_p$ (˚C) | In the first stage | | In the second stage | | | |
|---|---|---|---|---|---|---|---|---|
| | | | $T_{f1}$ (˚C) | $DTG_{max1}$ (%/min) | $T_{f2}$ (˚C) | $DTG_{max2}$ (%/min) | $T_{f3}$ (˚C) | $DTG_{max3}$ (%/min) |
| CG | 405.6 | 531.2 | - | - | 493.1 | 3.2 | - | - |
| PVC | 420.1 | 586.2 | 299.4 | 18.1 | 449.1 | 3.2 | 534.8 | 4.3 |

CG and PVC are coal gangue and polyvinyl chloride, $T_i$ is the ignition temperature, $T_p$ is the temperature corresponding to the maximum rate of weightlessness, $T_f$ is the burnout temperature, and $DTG_{max}$ is the maximum weightlessness rate

**Pyrolysis and combustion analysis.**

(1) Pyrolysis and combustion of CG

The TG-DTG-DSC curves of pyrolysis and combustion of CG in a $N_2$ and air atmosphere are shown in Fig 6. The temperature range of the pyrolysis reaction of CG was close to that of the combustion reaction, and weight loss occurred for both pyrolysis and combustion at an ignition temperature of 405.6˚C, indicating that the activation energy of pyrolysis is near that of combustion during the initial stage. Pyrolysis weightlessness is mainly due to the release of volatiles and gasification of liquid tar and gaseous substances such as $H_2$, $CH_4$, $H_2O$, and $CO_2$. Meanwhile, sulfur-containing compounds are produced during volatilization. During pyrolysis, different types of gases are released at different temperature ranges. The release of $H_2$ is mainly concentrated in the temperature range of 400–500˚C and at temperatures above

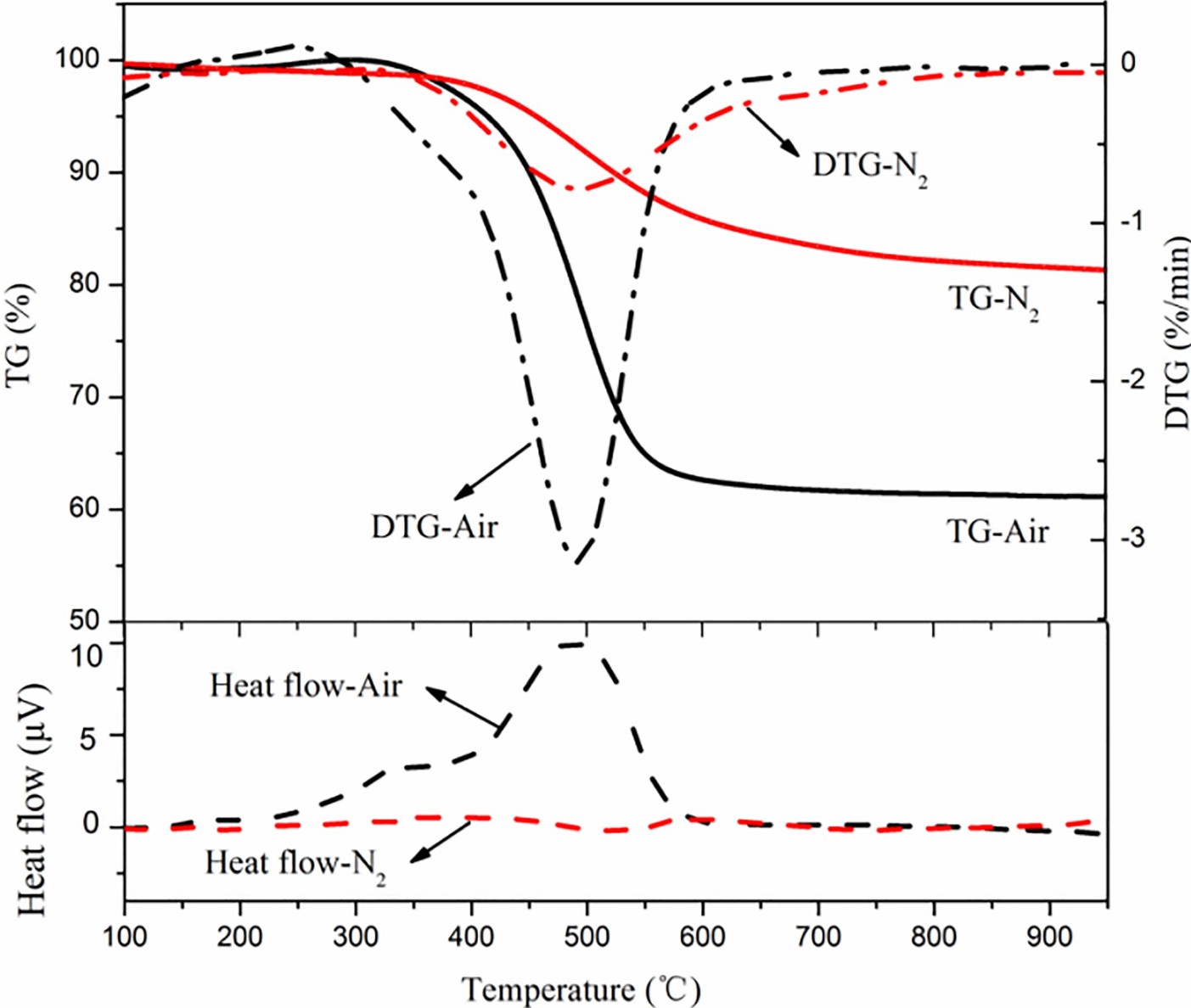

**Fig 6. Thermogravimetric–differential thermogravimetric–differential scanning calorimetry (TG-DTG-DSC) curves of combustion and pyrolysis of coal gangue.**

600˚C. In the range of 400–550˚C, the hydrogen-rich matrix is mainly degraded by heating. When the temperature is greater than 600˚C, the semi-coke condenses, the aromatic nuclei increases, the regularized arrangement of carbon structural units strengthen, a polycondensation reaction of aromatic and hydrogenated aromatic hydrocarbons occur, and the heterocyclic compounds decompose. $CH_4$ is generated in two ways. One is through the breaking of chains in the–$CH_3$ functional group and -$CH_2$ radical (>600˚C). The other is through the reaction of carbon and hydrogen from the splitting of ether or aromatic heterocyclic structures (>650˚C). The release of $H_2O$ is also separated into two stages: removal of free water from coal below 120˚C and formation of water after pyrolysis of partially dehydroxylated groups above 200˚C [46].

The weightlessness mass of CG in the combustion process was about 40%, while the weight loss before combustion was only 6%. The primary weightlessness stage occurred in the combustion stage, that is, the weightlessness mass of CG was about 33% in the temperature range of 405.6˚C to 531.2˚C. The $DTG_{max}$ value during combustion was 3.2%/min, and the temperature corresponding to $DTG_{max}$ was 493.1˚C, which corresponded to the temperature of the endothermic peak of the DSC curve. However, the average weightlessness rate was 1.24%/min, and the combustion rate was slow. Because the internal structure of CG is stable and the volatile content is low, the combustion is a gas-solid reaction dominated by fixed carbon. The total reaction rate is affected by the diffusion of oxygen. As the ash content in CG is rather high, the diffusion of oxygen is hindered, prolonging the reaction time in the burnout stage. As a result, the entire combustion process showed a mild weight loss. The $DTG_{max}$ value of pyrolysis was 0.8%/min below that of combustion, which corresponded to a temperature of approximately 500˚C. The pyrolysis process was slower and had a wider temperature range than the combustion process. The weightlessness of combustion can be regarded as the co-action of pyrolysis and fixed-carbon weightlessness; therefore, the average weightlessness rate of combustion is much higher than that of pyrolysis weightlessness. Above 580˚C, the weightlessness rate of combustion was lower than that of pyrolysis, and the combustion eventually reached equilibrium before pyrolysis. The figure shows that the combustion rate appeared to exhibit a constant weight at 600˚C. However, a slow mass loss for the pyrolysis lasted until the temperature reached the range of 800–900˚C. The final solid product was 83.7% of the sample for pyrolysis. Due to the different pyrolysis conditions affecting the formation of pyrolysis products, the weight loss mass is slightly larger than the volatile content of the proximate analysis of 16.3% (dry basis).

For the combustion process, the initial stage involved the absorption of a small amount of heat because of the release of only volatiles, while the other stages were exothermic. During combustion, the amount of emitted heat was far greater than the amount of absorbed heat, which was not apparent in the diagram. Compared with the combustion process, the entire pyrolysis process was endothermic.

(2) Pyrolysis and combustion of PVC

The TG-DTG-DSC curves of pyrolysis and combustion of CG in a $N_2$ and air atmosphere are shown in Fig 7. The pyrolysis process of PVC can be separated into two stages. In the first stage, the temperature range was 260–400˚C, $DTG_{max}$ was 13.4%/min where the weightlessness mass was fast, and $T_p$ was 315˚C. The temperature range of the second stage was 400–550˚C, and $T_p$ was 525˚C. The rate of weightlessness in the first stage is higher than that in the second stage because the irregular structures in the molecular chain, lower bond dissociation energy, and autocatalysis of Cl reduced the activation energy required for the reaction [47], such that the free radical Cl produced during the pyrolysis process absorbs H, forming the chain radical first and the double bond later. At the same time, HCl in a free state plays an autocatalytic role in the further removal of Cl in the molecular chain; hence, it belongs to the

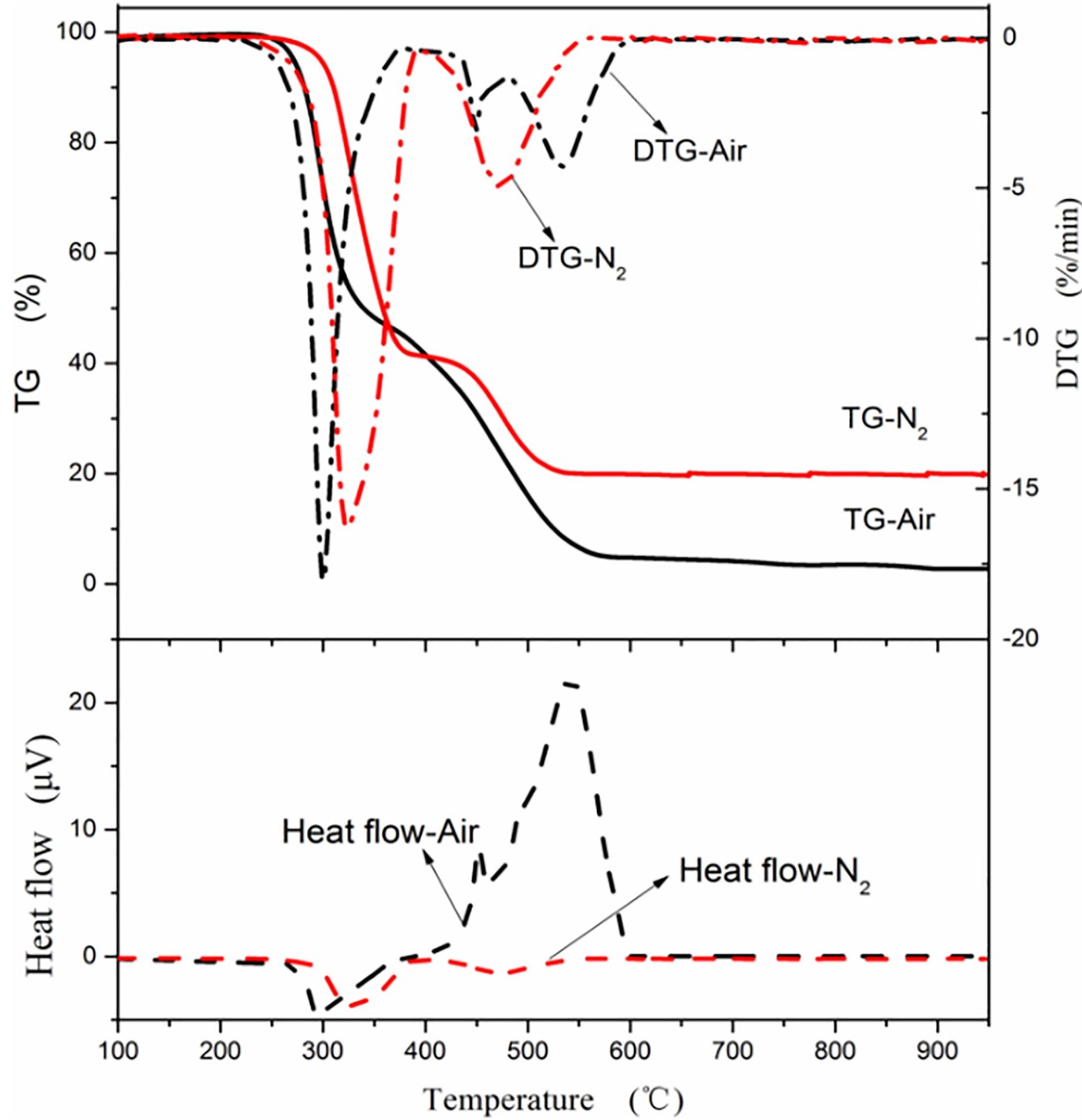

**Fig 7. Thermogravimetric–differential thermogravimetric–differential scanning calorimetry (TG-DTG-DSC) curve of combustion and pyrolysis of polyvinyl chloride.**

ionic reaction mechanism. With the continuous removal of HCl, the reaction rate decreases. If the temperature continues to rise, the polyolefin chain breaks down to form smaller olefin volatiles and tar. The resulting solid residue accounted for approximately 16% of the sample.

The combustion process of PVC can also be separated into two stages. In the first stage, the temperature range was 260–400˚C, $DTG_{max}$ was 18.1%/min, and $T_p$ was 300˚C. The emission data showed that HCl was basically released in this stage, which was very similar to the pyrolysis stage of PVC. In the initial process of combustion, pyrolysis occurs before the combustible is on fire. Because of the existence of oxygen, the pyrolysis process before ignition is accompanied by slow oxidation, which is reflected in the gradual increase in the amount of heat released in the DSC curve, and the temperature corresponding to $DTG_{max}$ is slightly lower than that of pure pyrolysis. The temperature range of the second stage was 410–560˚C, and $T_p$ was 525˚C. In the second stage, the residual carbon and hydrogen products from pyrolysis are oxidized, and more heat is released. It is worth noting that two consecutive peaks are formed in the DTG curve at this stage. It is presumed that two parallel reactions occurred and are related to the polyolefin chains and cyclic aromatic hydrocarbons formed in the first stage. At the end of combustion, the combustion residue was only 2.7% of the sample, which is mainly due to the low impurity content of the PVC sample.

**Combustion dynamics analysis.** In this section, the activation energy of CG and PVC are obtained. According to the TG curve at the three different heating rates, namely 5˚C/min, 10˚C/min, and 30˚C/min, a series of linear clusters can be obtained by the linear fitting of the calculated values of $1/T$ and $\ln(\beta/T^2)$, with a conversion ratio from 0.025 to 0.975 and an equal spacing of 0.05 as the step size. The results are shown in Fig 8. According to the method described in Section 2.4.2, the slope of each straight line in a cluster of straight lines is the ratio of the activation energy to the general gas constant at this conversion ratio. Thus, the corresponding activation energy can be obtained.

As can be seen from Fig 8, the obtained coordinate points at different heating rates had a high linear correlation, that is, the fitting degree is rather good; thus, the activation energy can be calculated accurately.

(1) Combustion dynamics analysis of CG

The values and change trends of the activation energy of CG at different conversion ratios are shown in Fig 9. It can be seen from Fig 9 that the change in activation energy of CG with conversion ratio had a "U"-trend distribution law, and the curve can be divided into three stages.

When the conversion ratio was less than 0.1, the reaction was in the initial stage. With the increase in conversion ratio, the activation energy decreased rapidly to 132.6 kJ/mol, and the average activation energy was 143.2 kJ/mol. This may be related to the chemisorption of CG in the initial stage of combustion and the high combustion ratio of volatiles in the initial stage of combustion. The decrease in activation energy is related to the catalytic combustion of the minerals in ash. The composition and structure of minerals, especially clay minerals, play an important role in the initial release of volatiles.

When the conversion ratio was greater than 0.1 and less than 0.81, the activation energy curve appeared as a straight line inclined downward, and the fixed carbon combustion occurred mainly at this stage. With the increase in temperature, the release rate of volatiles decreased considerably, the originally dispersed mineral particles tended to aggregate gradually, and the catalytic action of minerals disappeared. When the conversion ratio increased to 0.81, the activation energy decreased to 94.2 kJ/mol. The composition of pyrite in CG can promote the oxidation of CG, and the increase in pore structure during combustion can increase the contact between oxygen and the combustible material [48].

When the conversion ratio was greater than 0.81, the activation energy increased rapidly again. This was caused by the reduction in fixed carbon during combustion, the increase in ash content, and the endothermic decomposition of stable minerals, such as dolomite, at high temperatures [49].

(2) Combustion dynamics analysis of PVC

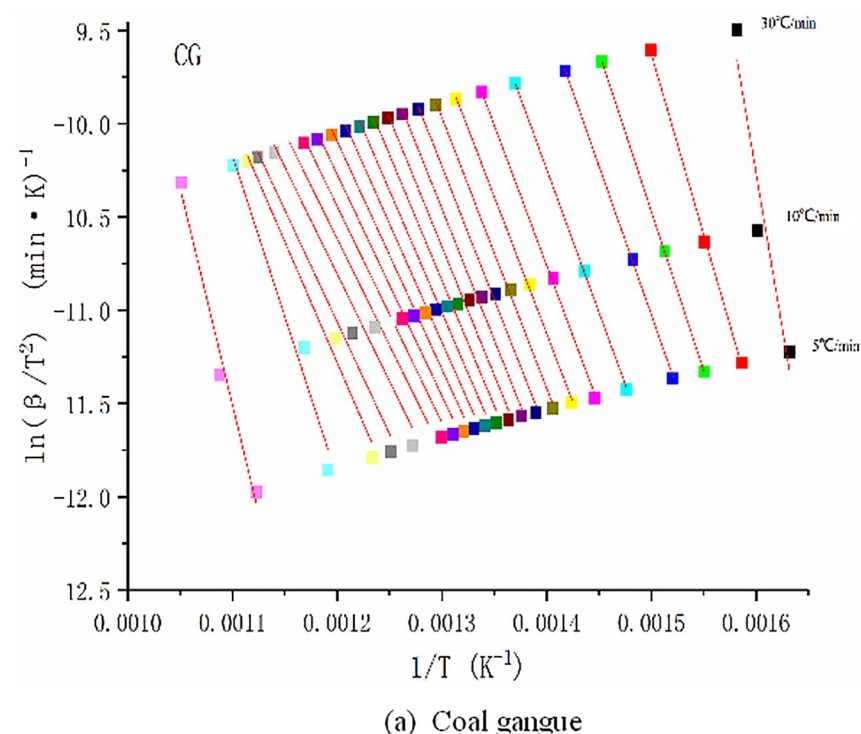

(a) Coal gangue

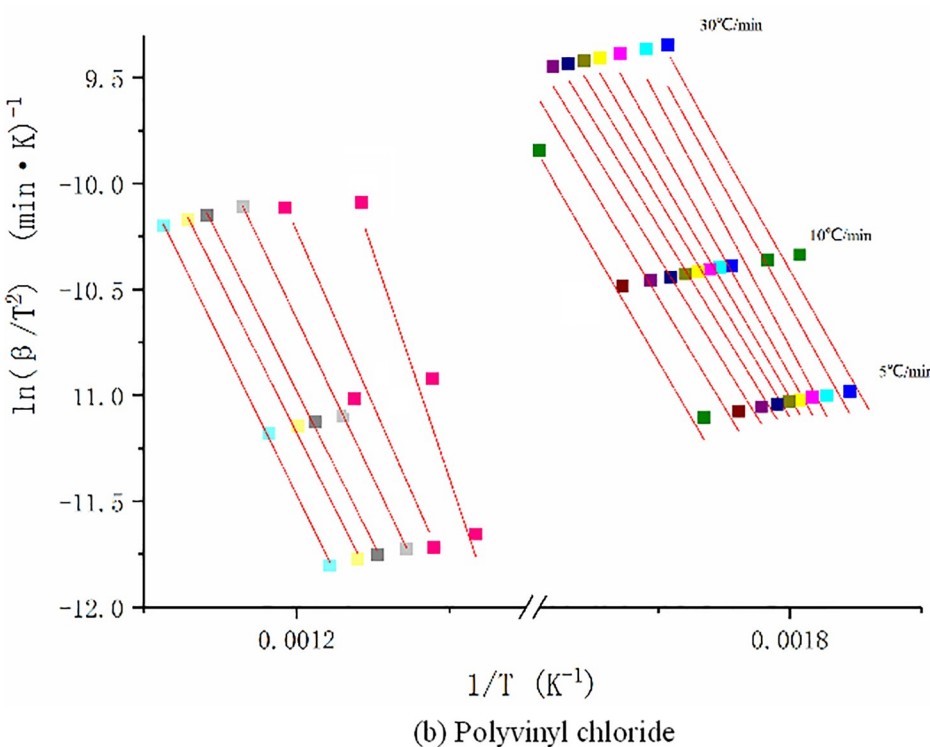

(b) Polyvinyl chloride

**Fig 8.** Fitting curves of kinetics of (a) coal gangue and (b) polyvinyl chloride.

The values and change trends of the activation energy of PVC at different conversion ratios are shown in Fig 10. From Fig 10, it can be seen that the change in activation energy of PVC with conversion ratio follows an "M"-trend distribution.

At a conversion ratio of less than 0.5, the activation energy decreased slowly from 100 kJ/mol to 85 kJ/mol. With further removal of Cl in the molecular chain, the autocatalytic role of free-state HCl resulted in a decrease in activation energy. The average activation energy of the initial reaction stage with a conversion rate of less than 0.1 was 93.4 kJ/mol.

When the conversion ratio was greater than 0.5, the activation energy increased first and then decreased. The activation energy increased to 185 kJ/mol because of the polyolefins and aromatic hydrocarbons produced by the reaction. When the temperature continued to rise, the activation energy decreased again after the breaking of the chains of polyolefins and cracking of aromatic hydrocarbons.

### Experimental results and discussion on co-combustion

**Experimental results of co-combustion.** Fig 11 shows the TG curves of the combustion of CG and PVC, and the co-combustion of CG with PVC in different proportions. With the

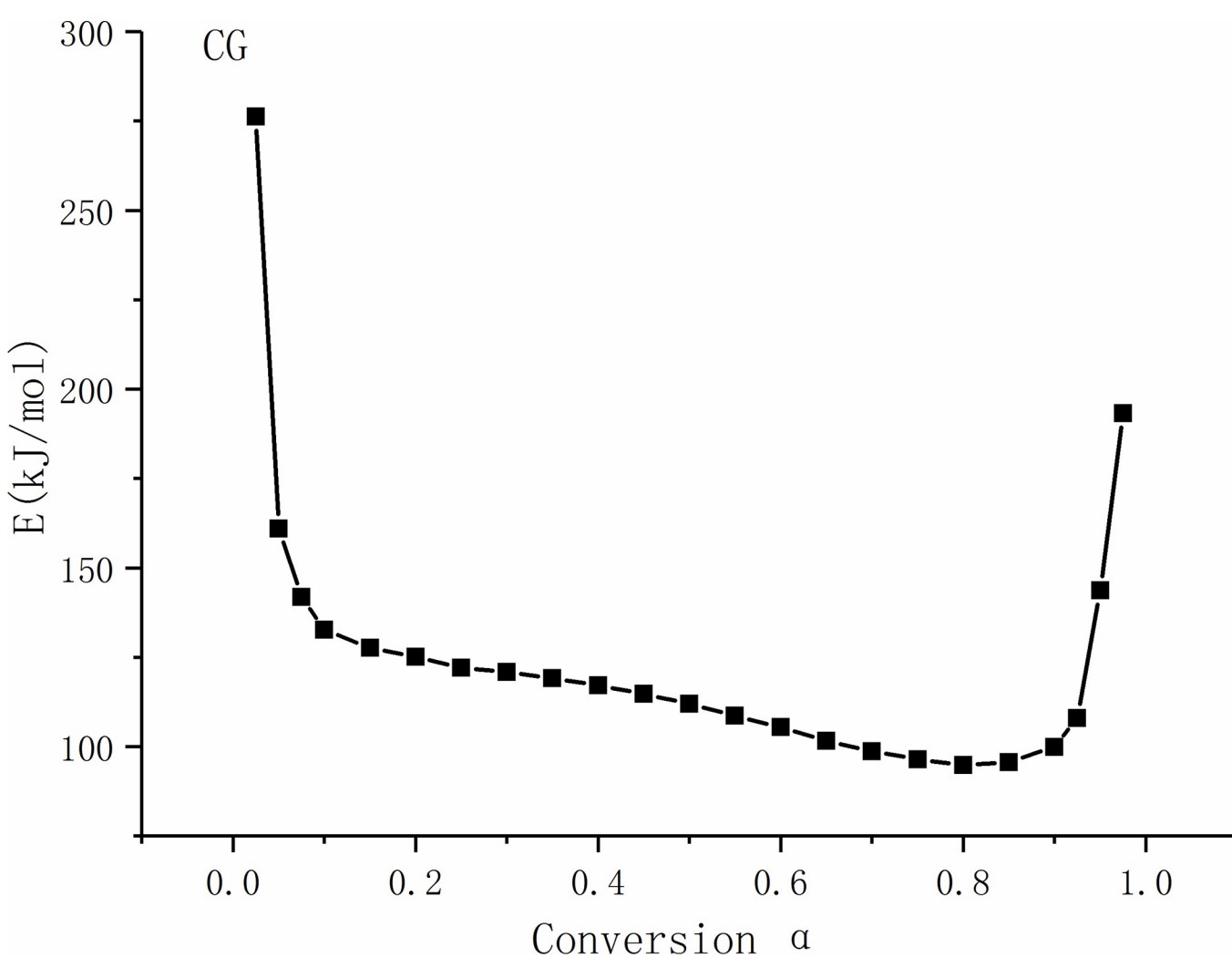

**Fig 9. Activation energy curve of coal gangue with different conversion ratios.**

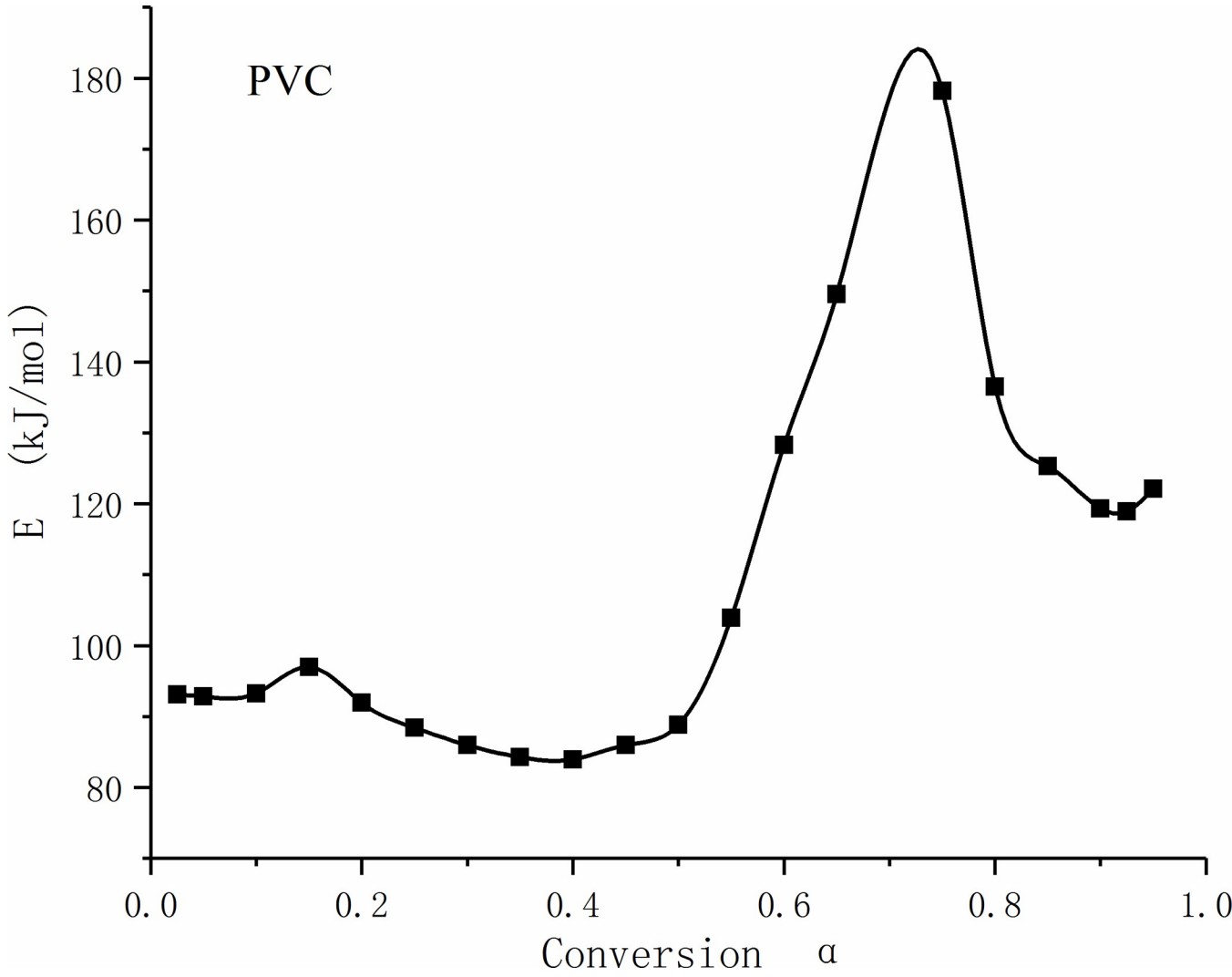

**Fig 10. Activation energy curve of polyvinyl chloride with different conversion ratios.**

increase in the ratio of PVC, the weightlessness curve gradually moved down and finally increased gradually in the rapid weightlessness stage. There were three combustion weightlessness peaks when PVC was burned alone. However, there were only two combustion weightlessness peaks when PVC was mixed with gangue and burned together. According to the DSC curves of CG and PVC combustion, which is shown in Fig 4, PVC was in a violent endothermic pyrolysis stage at approximately 300°C, while CG was in the slow oxidation and exothermic stage of the hydrocarbon radical in the volatilization. Although CG is not on fire at this time, the heat emitted by CG can significantly accelerate the combustion due to the weightlessness of PVC. Because the maximum rate of weightlessness in the second stage of PVC is exothermic and the released heat is greater than the released heat of CG at $T_p$ in the second stage, the weightlessness rate of CG is accelerated after heat absorption, and the weightlessness rate of PVC is reduced after heat loss. Therefore, the $DTG_{max}$ peak at this stage disappears.

Table 5 shows the combustion characteristic parameters of CG and PVC, and the co-combustion of CG with PVC in different proportions. The final weightlessness mass and the

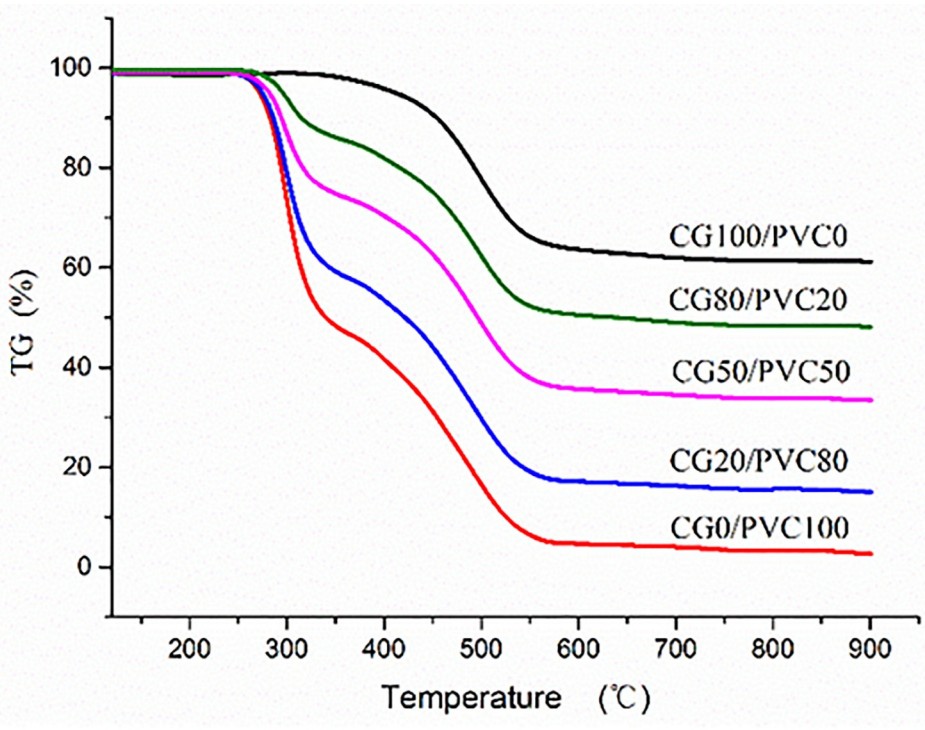

(a) Thermogravimetric curves

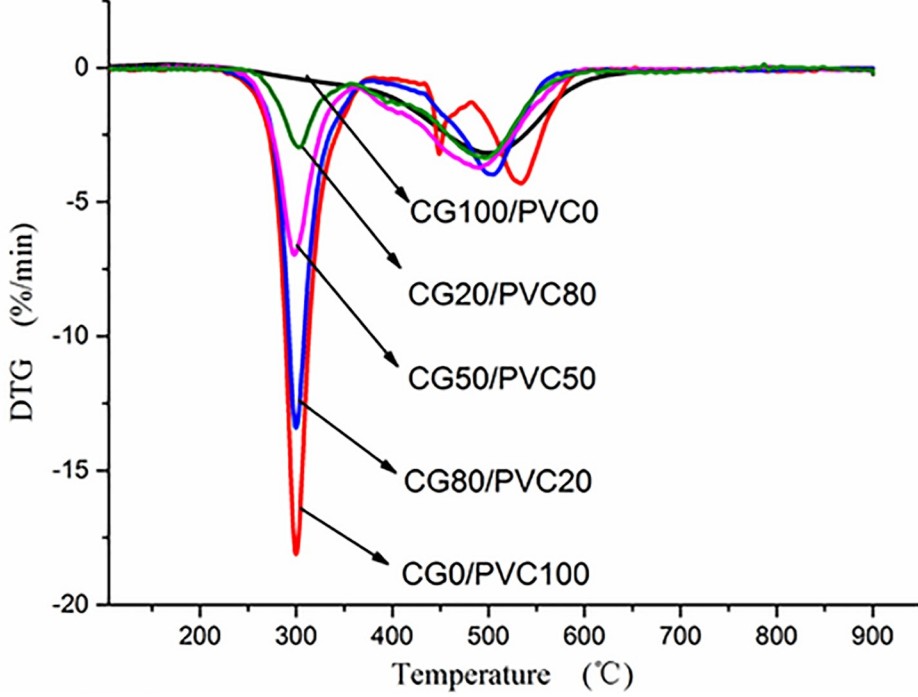

(b) Differential thermogravimetric curves

**Fig 11.** (a) Thermogravimetric and (b) differential thermogravimetric curves of co-combustion of coal gangue (CG), polyvinyl chloride (PVC), and CG and PVC mixtures with different proportions.

**Table 5. Characteristic parameters upon co-combustion of coal gangue and polyvinyl chloride.**

| Percentage of CG(%) | | 100 | 80 | 50 | 20 | 0 |
|---|---|---|---|---|---|---|
| Percentage of PVC (%) | | 0 | 20 | 50 | 80 | 100 |
| $T_i$ (°C) | | 405.6 | 408.4 | 412.9 | 418.4 | 420.1 |
| $T_f$ (°C) | | 531.2 | 534.2 | 542.1 | 560.6 | 586.2 |
| $TG_{max}$ (/%) | | 38.4 | 50.8 | 68.0 | 86.2 | 97.3 |
| In the first stage | $T_{p1}$ (°C) | - | 303.6 | 297.7 | 293.7 | 299.4 |
| | $DTG_{max1}$ (%/min) | - | 4.5 | 10.3 | 15.6 | 18.1 |
| In the second stage | $T_{p2}$ (°C) | - | - | - | - | 449.1 |
| | $DTG_{max2}$ (%/min) | - | - | - | - | 3.2 |
| | $T_{p3}$ (°C) | 493.1 | 495.6 | 494.3 | 503.5 | 534.8 |
| | $DTG_{max3}$ (%/min) | 3.2 | 3.7 | 4.0 | 4.2 | 4.3 |

$T_i$ is the ignition temperature, $T_p$ is the temperature corresponding to the maximum rate of weightlessness, $T_f$ the burnout temperature, $TG_{max}$ is the maximum weightlessness value, and $DTG_{max}$ is the maximum weightlessness rate

$DTG_{max}$ of PVC were much higher than those of CG. Thus, PVC is easier to burnout and more intense to burn than CG is.

**Characteristic temperature parameter analysis.** The variations in the ignition temperature and burnout temperature at different mixing proportions of CG and PVC are shown in Fig 12. With the increase in the PVC ratio, the ignition temperature decreased because the ignition temperature of PVC was lower. The burnout temperature increased with an increase in the PVC ratio, because the supporting effect of loose ash formed by CG combustion was reduced, leading to an increase in oxygen diffusion resistance.

The effect of mixing ratio on $T_p$ is shown in Fig 13. The $T_p$ value for the co-combustion process is related to the proportion of PVC in the mixture. At approximately 300°C, both CG and PVC were in the endothermic pyrolysis stage, and the reaction of HCl gas from PVC pyrolysis with mineral salts in the unburned CG was very weak. Although the reaction of mineral salts with HCl was exothermic, the release of heat could be neglected compared with the absorption of heat of CG and PVC in pyrolysis. Thus, when the proportion of PVC increased, $T_p$ of the co-combustion quickly approached the $T_p$ of PVC combustion alone, and the value of $T_p$ first decreased slightly and then increased. At $T_p$ in the third stage, the CG unburned material covered the PVC unburnt material (the amount of which was less than that of CG), hindering the contact between PVC and air to a certain extent and resulting in the shift of $T_p$ to the high-temperature zone.

**Correlation analysis of co-combustion.** In this section, the TG experimental curves three proportions of CG and PVC blends are compared with those of CG and PVC. To further analyze the interaction between CG and PVC during co-combustion, the experimental TG curves at three mixing ratios were compared to the theoretically calculated curves obtained by the linear superposition of the individual TG curves of CG and PVC. The results are shown in Fig 14. The initial and end stages of the theoretically calculated curves for different proportions were coincident with that of the experimental TG curves, that is, the calculated maximum weightlessness mass was the same as the actual weightlessness mass. This indicates that the final solid residuals from the co-combustion of CG with PVC are linear combinations of solid residuals from their separate combustion. In this study, the ash formed in the mixed combustion was compared with that formed during the combustion of CG and PVC alone, and the results showed that there was no change in the ash produced during co-combustion. Therefore, it is inferred that there was no chemical reaction in the co-combustion of the two, but rather the

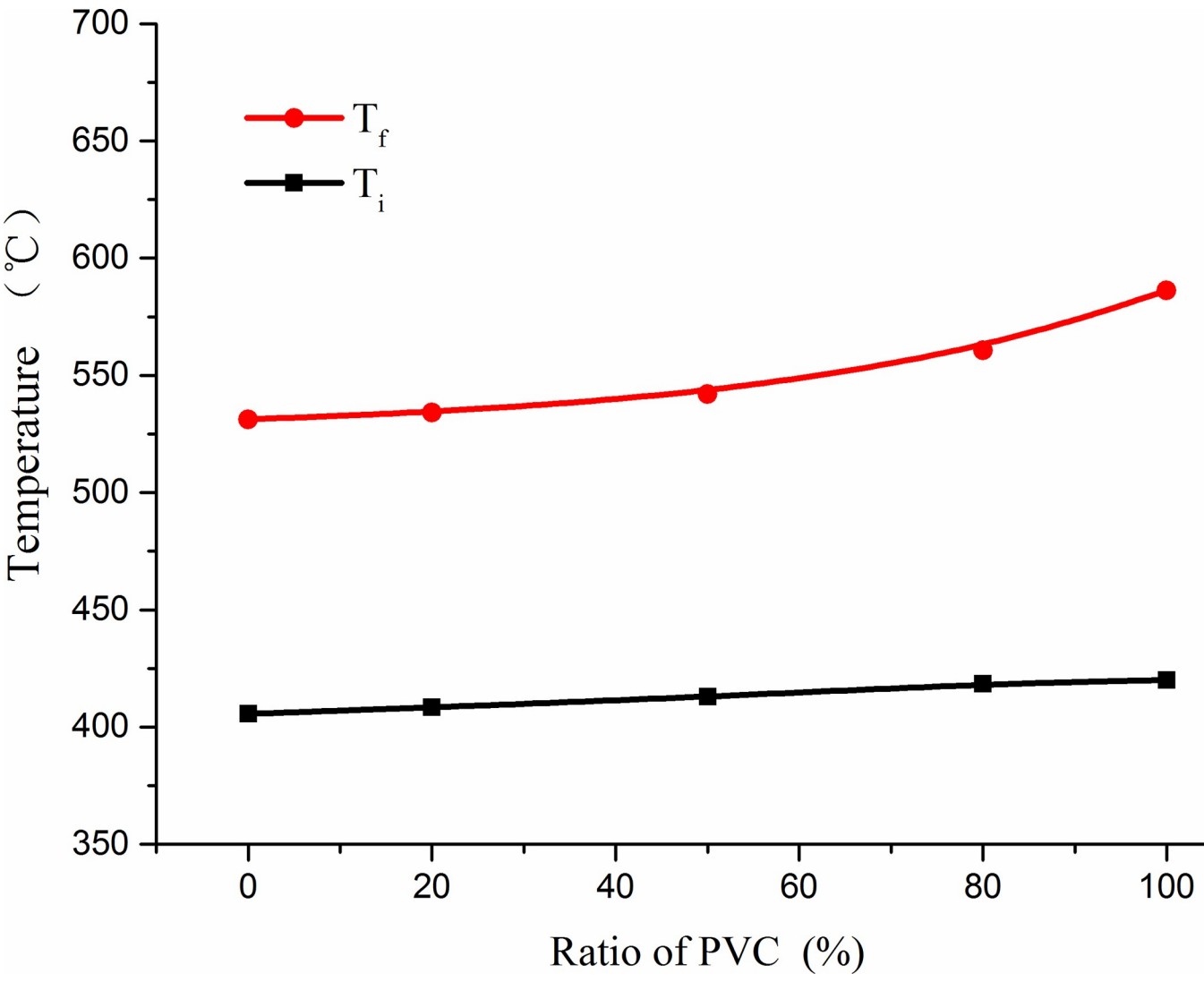

**Fig 12. Effect of coal gangue and polyvinyl chloride (PVC) mixing ratio on ignition temperature ($T_i$) and burnout temperature ($T_f$).**

interaction was only thermal coupling. It can be seen from Fig 14 that the experimental TG curves are all below the theoretically calculated TG curves, and the separation degree of the two lines was the most significant when the ratio of PVC to CG was 20%:80%. To obtain the characteristics of the co-combustion, the change trend of $DTG_{max}$ in the co-combustion of PVC with CG was plotted.

As shown in Fig 15, the experimental maximum weightlessness rates for the three proportions were compared with the calculated maximum weightlessness rate. The $DTG_{max}$ of CG was calculated according to the linear sum of the weightlessness rate of CG and PVC burning separately at a temperature corresponding to $DTG_{max}$ during mixed combustion. The experimental maximum weightlessness rates increased with increase in the PVC ratio, but the weightlessness rate at low temperatures increased faster than that at high temperatures. The combustion rate during mixed combustion was larger than that during single combustion, that is to say, the combustion of the two mixtures play a better role in promoting combustion.

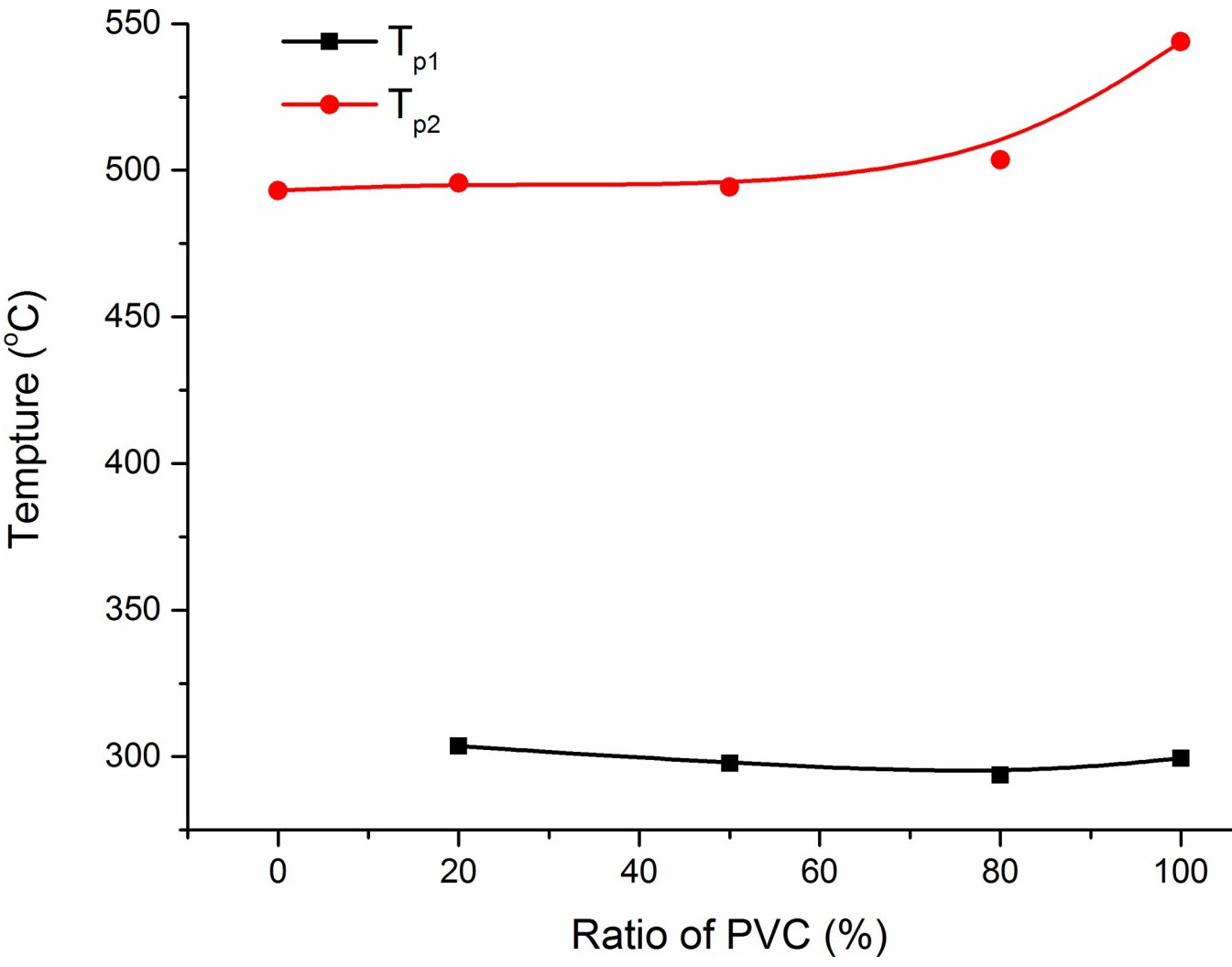

**Fig 13. Effect of coal gangue and polyvinyl chloride (PVC) mixing ratio on temperature corresponding to the maximum rate of weightlessness ($T_p$).**

The positive coupling effect can be seen more clearly from a comparison of the experimental values and calculated values of the co-combustion, as shown in Table 6. When PVC and CG are mixed, the combustion increases at different ratios and different peak temperatures. The coupling degree at a low temperature corresponding to the maximum rate of weightlessness is stronger than that at a high temperature corresponding to the maximum rate of weightlessness, and the relative deviation for 80%:20% is the largest of the three proportions of mixed samples, that is, the coupling is the strongest for this proportion.

As mentioned earlier, PVC was in a thermal desorption process at temperatures below 300°C, while CG was in a slow oxidation exothermic process. The absorption of heat from CG by PVC greatly increased the weightlessness process of PVC during the co-combustion of the two substances, demonstrating that combustion was promoted. Because combustion is promoted, $T_p$ decreases. Because the heat released from CG is less than the heat required for PVC pyrolysis, the heat released from CG can satisfy the heat needed for the pyrolysis of PVC only when CG accounts for a large proportion. Thus, the ratio of CG to PVC of 80%:20% produced the greatest heat transfer among the three proportions, and the weightlessness rate of PVC

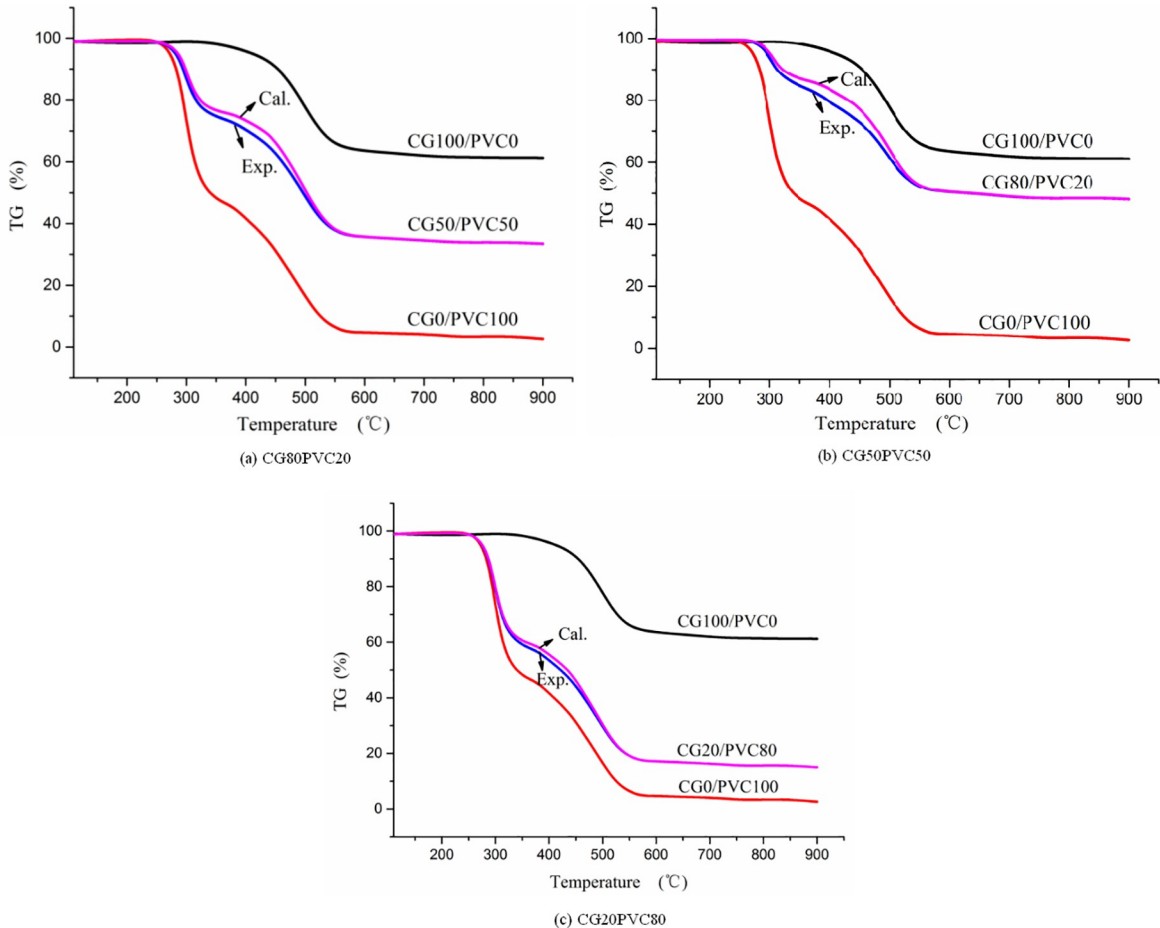

**Fig 14.** Comparison of experimental (Exp.) and calculated (Cal.) thermogravimetric (TG) of different coal gangue (CG) and polyvinyl chloride (PVC) mixing ratios: (a) CG80PVC20, (b) CG50PVC50, and (c) CG20PVC80.

near this temperature was much larger than that of CG. That is to say, the promoting effect is most significant when the ratio of CG to PVC is 80%:20%.

According to Section 3.1.2, both CG and PVC were in the exothermic phase when they burned alone and at $T_p$ in the second stage. There were more solid-state residual products, when CG burned out. The large amounts of ash content from burning CG can play a supporting role for PVC, making it easier for air to flow through it and accelerate its combustion. At this time, the combustion of PVC can release more heat and promote the combustion of CG. Because both are exothermic and there is little difference in the amount of released heat at this moment, the coupling is weaker in this stage than in the first stage. The more the amount of CG in the mixture, the stronger the supporting effect on PVC in the mixture, and the more it can promote the combustion of PVC.

**Numerical analysis of co-combustion interaction.** In this section, the coupling of the co-combustion is obtained by analyzing the value and properties of the interaction. According to the previous analysis, the curves of λ of the co-combustion of PVC with CG, which is shown in Fig 16, were obtained using Eq (14). The value of $\bar{\lambda}$ and the properties of the interaction between the two mixtures were calculated as shown in Table 7.

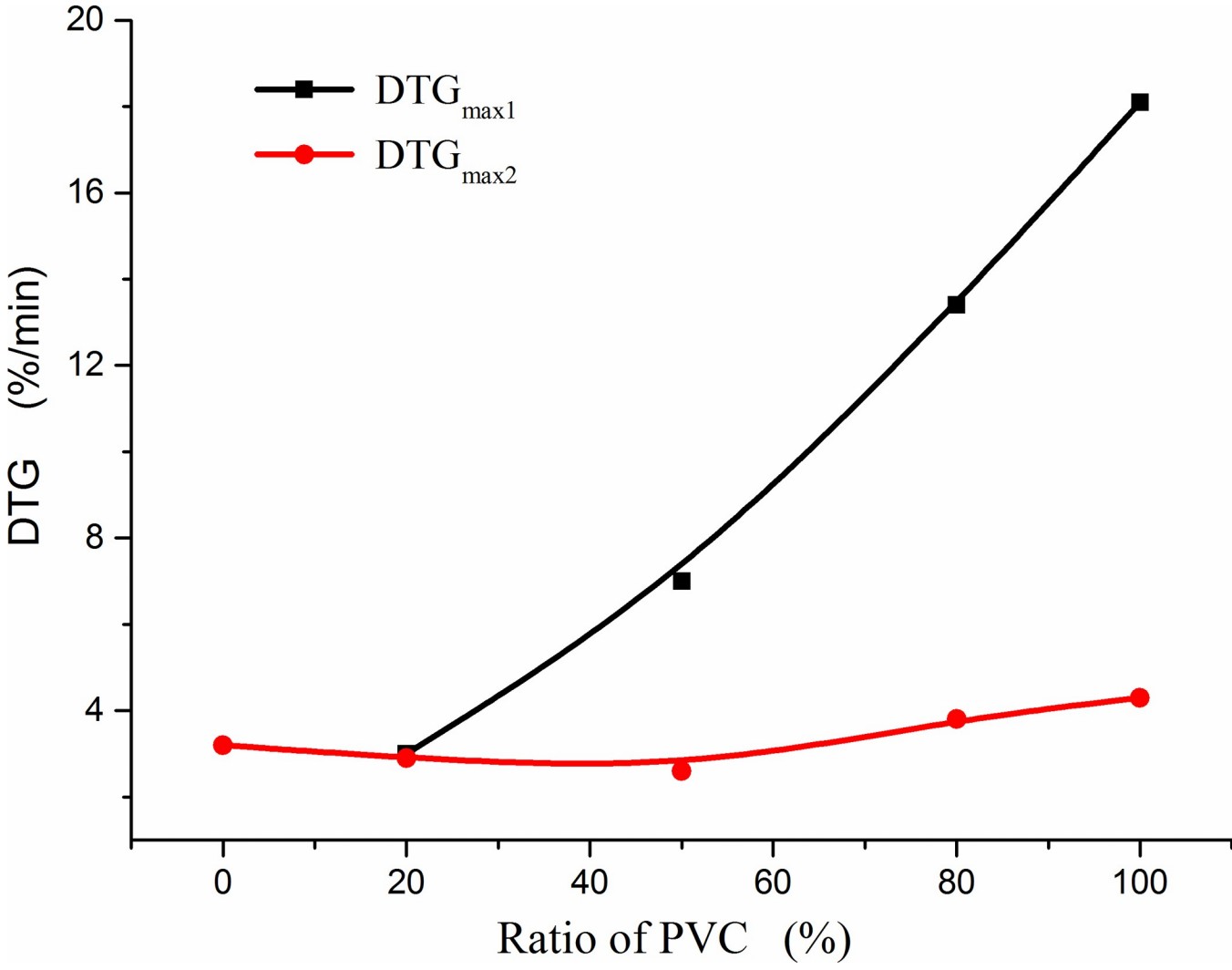

**Fig 15. Effect of co-combustion of coal gangue and polyvinyl chloride (PVC) mixtures with different mixing proportions on maximum weightlessness rate ($DTG_{max}$).**

As can be seen from Fig 16, there was a similar trend in $\lambda$ in the mixed combustion of CG and PVC in different proportions. Consistent with the results of the previous analysis, the mixture of the two promotes the combustion of CG. The coupling effect increased from 300˚C to the maximum and then decreased to zero at a faster speed.

It can be seen from the above calculations that the maximum values of $\lambda$ for the three ratios were 0.00318, 0.00286, and 0.00236. The maximum value of the temperature-varying co-

**Table 6. Comparison of experimental and calculated data for co-combustion of coal gangue (CG) and polyvinyl chloride (PVC).**

| Classification | In the first stage | | | In the second stage | | |
|---|---|---|---|---|---|---|
| Mixing ratio (CG%:PVC%) | 80:20 | 50:50 | 20:80 | 80:20 | 50:50 | 20:80 |
| Experimental values (%/min) | 4.53 | 10.30 | 15.60 | 3.74 | 4.00 | 4.24 |
| Calculated values (%/min) | 3.94 | 9.25 | 14.56 | 3.42 | 3.75 | 4.08 |
| Maximum relative deviation (%) | 13.1 | 10.2 | 6.7 | 8.6 | 6.2 | 3.9 |
| Corresponding temperature (˚C) | 303.6 | 297.7 | 293.7 | 495.6 | 494.3 | 503.5 |

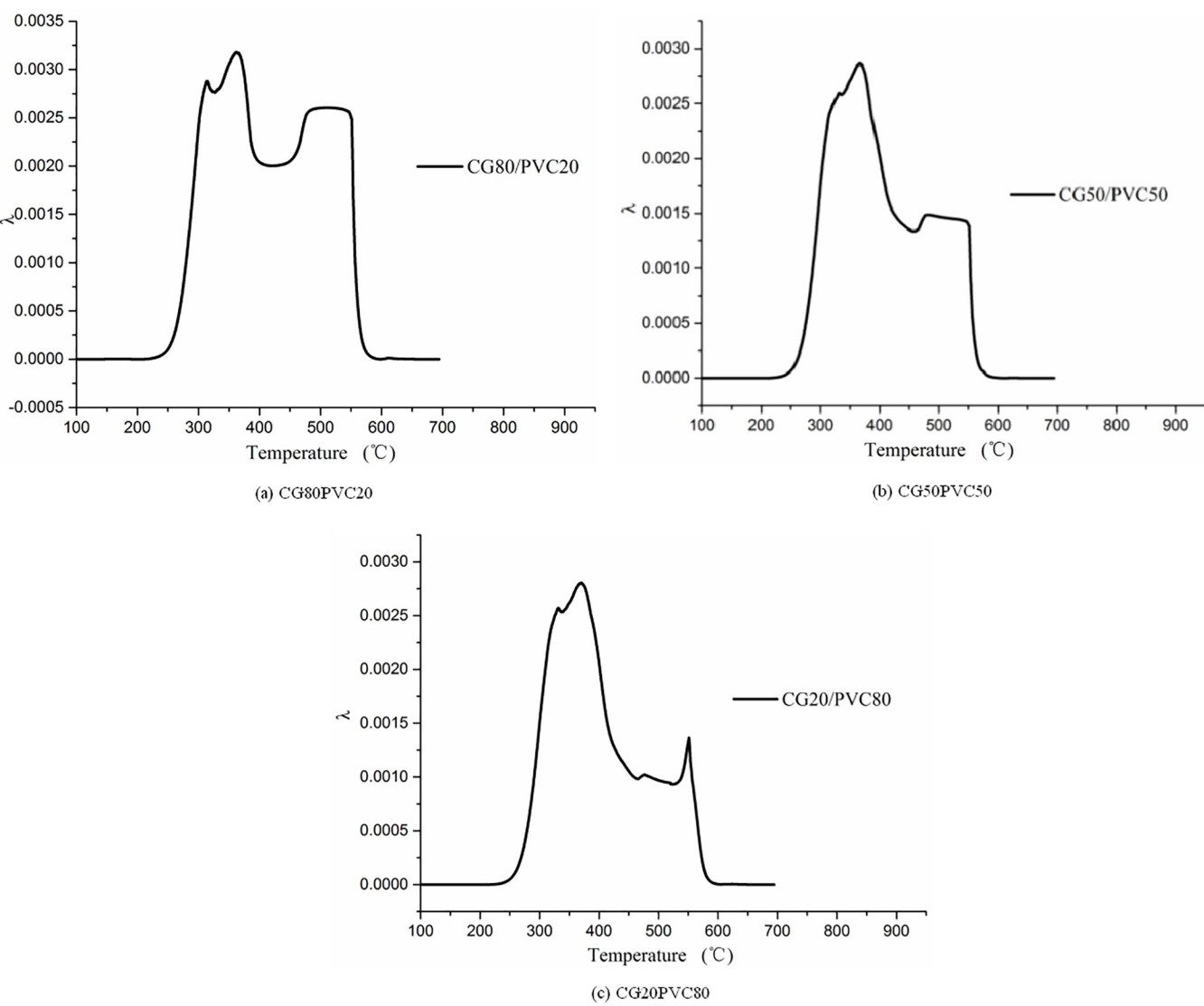

**Fig 16. Coupling change on co-combustion of coal gangue (CG) and polyvinyl chloride (PVC) mixtures of different proportions.**

combustion thermal coupling coefficient of 20% PVC is 1.11 times that of 50% PVC and 1.35 times that of 80% PVC. The values of $\bar{\lambda}$ were 0.0003975, 0.0003497, and 0.0002913. These results also indicate that the coupling interaction of co-combustion is the strongest when the proportion of PVC is 20% and the corresponding temperature is 300˚C. It can be seen that the coupling analysis method presented in this paper for co-combustion is effective for analyzing the interaction between PVC and CG during co-combustion.

**Table 7. Numerical calculation of coupling during co-combustion.**

| Classification | Mixing ratio | $\bar{\lambda}$ $(10^{-4})$ | $\Pi$ | Properties of interaction |
|---|---|---|---|---|
| CG:PVC | 80%:20% | 3.975 | 0.6957 | Promoting combustion |
| | 50%:50% | 3.497 | 0.5845 | Promoting combustion |
| | 20%:80% | 2.913 | 0.5281 | Promoting combustion |

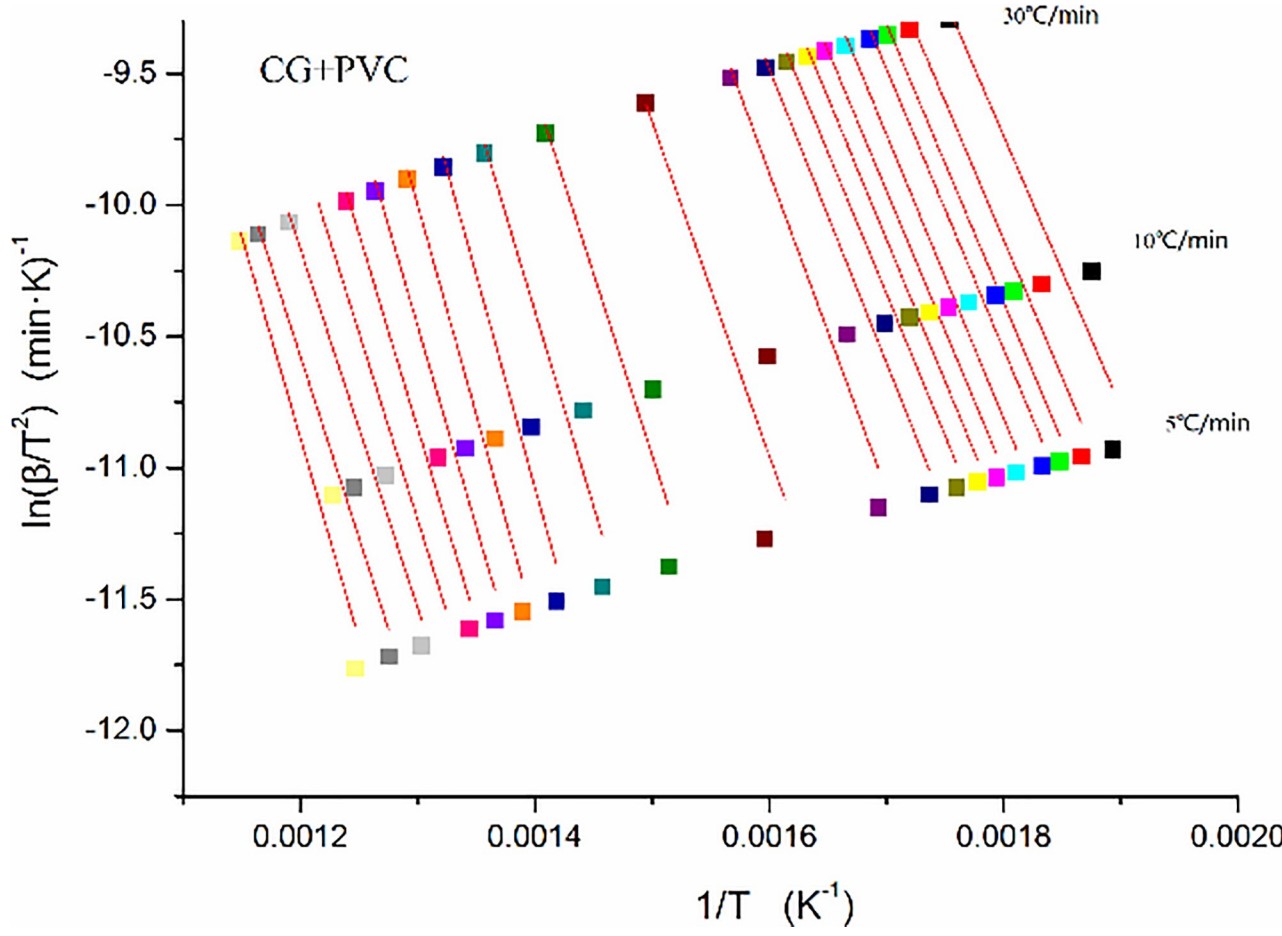

**Fig 17. Activation energy fitting curves of 50%:50% coal gangue (CG) and polyvinyl chloride (PVC) mixture.**

**Combustion dynamics analysis.** In this section, the activation energy of the co-combustion of CG and PVC in the same proportion are obtained. By using the same calculation method as that in Section 3.1.6, a series of linear clusters of the co-combustion of PVC with CG in the ratio of 50%:50% can be obtained. The results are shown in Fig 17. As mentioned earlier, the slope of each straight line in the linear cluster is the ratio of the activation energy at this conversion to the universal gas constant, and the corresponding activation energy is further obtained.

Fig 18 shows the variation tendency of the activation energy of CG, PVC, and the mixture of CG and PVC in the ratio 50%:50% at different conversion rates.

When the conversion ratio was less than 0.3, HCl was produced by the pyrolysis of PVC in the mixed fuel. The variation of the activation energy of the mixture was consistent with that of PVC, and the activation energy of PVC remained at approximately 94 kJ/mol during this stage. When the conversion ratio was less than 0.1, the reaction was in the initial stage, and the average activation energy was 89.8 kJ/mol.

When the conversion ratio was greater than 0.3 and less than 0.9, the activation energy showed a maximum value of 139 kJ/mol, which corresponded to a conversion rate of 0.62. This variation was caused by changes in the formation and cracking of polyene and aromatic hydrocarbons in PVC.

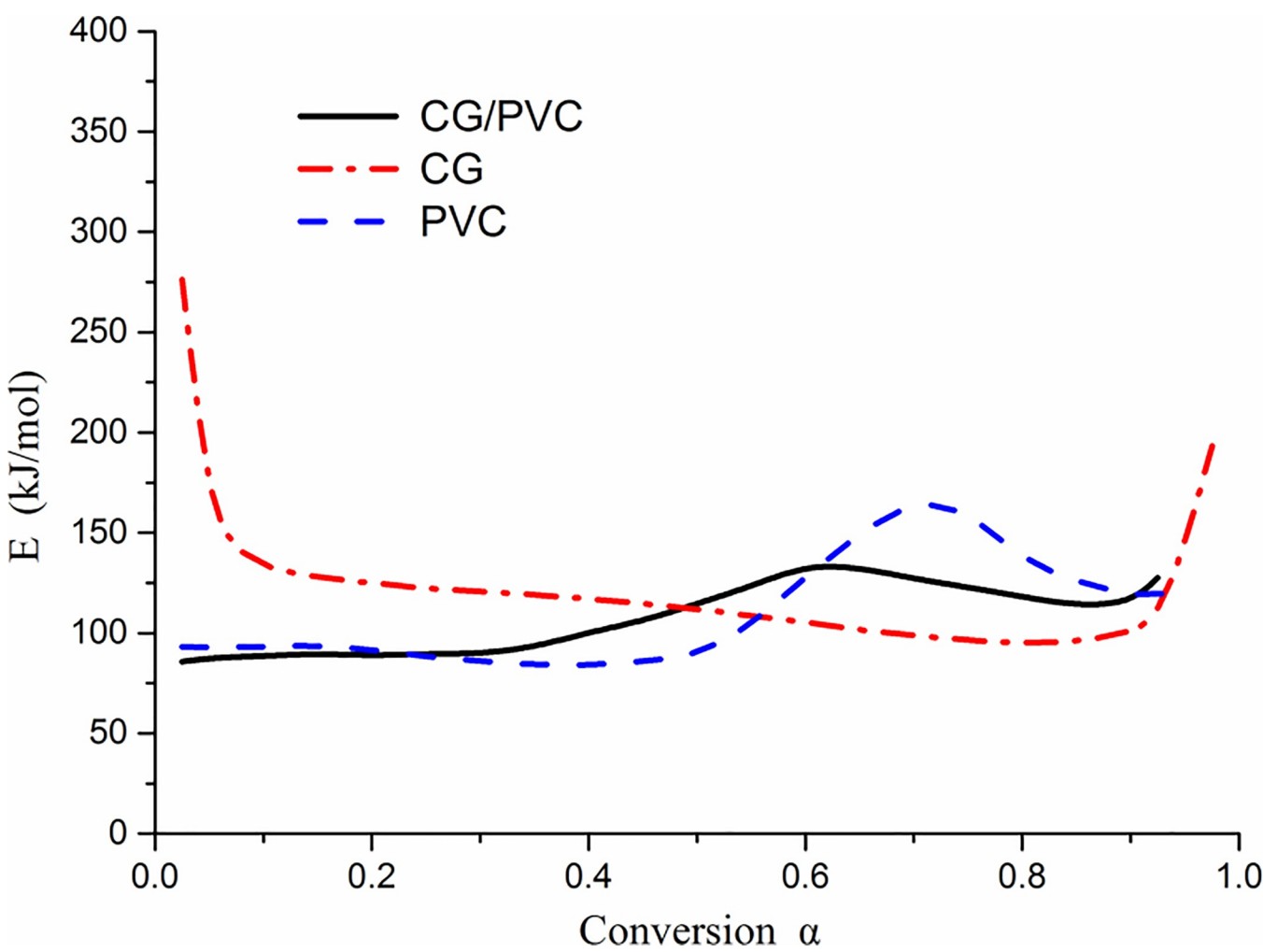

**Fig 18. Activation energy curves of coal gangue (CG), polyvinyl chloride (PVC), and the mixtures with different conversion ratios.**

When the conversion ratio was greater than 0.9, the activation energy of the mixture showed an increasing tendency. At this stage, the combustible composition of the mixture was lower, and the conversion rate continued to increase due to the weight loss of the more stable minerals, such as dolomite in CG, thereby producing a gradual increase in the activation energy of the mixture.

To summarize, on comparing the variation of the activation energy of co-combustion with that of individual components, it can be seen that the activation energy of CG at the beginning and the end of combustion can be greatly reduced by co-combustion. Through combustion by mixing with PVC, the difficulty in the combustion of CG alone is alleviated, and the activation energy of PVC in the later stages of combustion is also reduced. Therefore, co-combustion is helpful in improving the combustion performance of CG and PVC.

## Conclusions

In this paper, a treatment method based on the co-combustion of CG with PVC, which can be centrally recovered from municipal solid waste, was proposed. The TG experimental results of

CG, PVC, and the mixture of CG and PVC with different proportions were analyzed, and the main conclusions are summarized as follows.

1. The weightlessness curves and the weightlessness rate curves of CG and PVC shift toward the high-temperature region after the heating rate is increased.

2. The coupling analysis method presented in this paper can intuitively evaluate the interaction between PVC and CG during co-combustion.

3. There were strong interactions between CG and PVC during their co-combustion. With the increase in the PVC ratio in the mixture, the temperature corresponding to the maximum rate of weightlessness in the first stage first decreased slightly and then increased, while that in the second stage kept increasing. The maximum weightlessness rate in the first stage exhibited a rapid increase, while the maximum weightlessness rate in the second stage exhibited a slow increase. The promotion effect was the most significant with 20% PVC.

4. CG and PVC exhibited very different combustion behaviors. For 20% PVC, the co-combustion coupling was the strongest, and the maximum value of the temperature-varying co-combustion thermal coupling coefficient of 20% PVC was 1.11 times greater than that for 50% PVC and 1.35 times greater than that for 80% PVC.

5. Co-combustion is helpful to improve the problems of low calorific value and refractory burnout of coal gangue.

## Supporting information

**S1 Fig. Calibration results of experimental system.**
(XLS)

**S2 Fig. Thermogravimetry curves of combustion and pyrolysis for coal gangue (CG) and polyvinyl chloride (PVC) in nitrogen.**
(XLS)

**S3 Fig. Thermogravimetry curves of combustion and pyrolysis for coal gangue (CG) and polyvinyl chloride (PVC) in air.**
(XLS)

**S4 Fig. Differential scanning calorimetry curves of flues in air atmosphere; CG and PVC are coal gangue and polyvinyl chloride.**
(XLS)

**S5 Fig. Effect of heating rate on single group combustion-Differential thermogravimetric curves of coal gangue.**
(XLS)

**S6 Fig. Effect of heating rate on single group combustion-Differential thermogravimetric curves of polyvinyl chloride.**
(XLS)

**S7 Fig. TG-DTG-DSC curves of combustion and pyrolysis of coal gangue.**
(XLS)

**S8 Fig. TG-DTG-DSC curve of combustion and pyrolysis of polyvinyl.**
(XLS)

**S9 Fig. Fitting curves of kinetics of coal gangue.**
(XLS)

**S10 Fig. Fitting curves of kinetics of polyvinyl chloride.**
(XLS)

**S11 Fig. Activation energy curve of coal gangue with different conversion ratios.**
(XLS)

**S12 Fig. Activation energy curve of polyvinyl chloride with different conversion ratios.**
(XLS)

**S13 Fig. Thermogravimetric curves of co-combustion of coal gangue (CG), polyvinyl chloride (PVC).**
(XLS)

**S14 Fig. Differential thermogravimetric curves of co-combustion of coal gangue (CG), polyvinyl chloride (PVC).**
(XLS)

**S15 Fig. Effect of coal gangue and polyvinyl chloride mixing ratio on ignition temperature (Ti) and burnout temperature (Tf).**
(XLS)

**S16 Fig. Effect of CG and PVC mixing ratio on temperature corresponding to the maximum rate of weightlessness.**
(XLS)

**S17 Fig. Comparison of experimental and calculated thermogravimetric of CG80PVC20.**
(XLS)

**S18 Fig. Comparison of experimental and calculated thermogravimetric of CG50PVC50.**
(XLS)

**S19 Fig. Comparison of experimental and calculated thermogravimetric of CG20PVC80.**
(XLS)

**S20 Fig. Effect of co-combustion of coal gangue and polyvinyl chloride (PVC) mixtures with different mixing proportions on maximum weightlessness rate ($DTG_{max}$).**
(XLS)

**S21 Fig. Coupling change on co-combustion of coal gangue (CG) and polyvinyl chloride (PVC) mixtures of CG80PVC20.**
(XLS)

**S22 Fig. Coupling change on co-combustion of coal gangue (CG) and polyvinyl chloride (PVC) mixtures of CG50PVC50.**
(XLS)

**S23 Fig. Coupling change on co-combustion of coal gangue (CG) and polyvinyl chloride (PVC) mixtures of CG20PVC80.**
(XLS)

**S24 Fig. Activation energy fitting curves of CG50PVC50.**
(XLS)

**S25 Fig. Activation energy curves of coal gangue polyvinyl chloride, and the mixtures with different conversion ratios.**
(XLS)

## Acknowledgments

The authors thank the support of the Excellent Talents Scientific and Technological Innovation Foundation of Shanxi Province (No. 201805D211039). We would like to thank Editage (www.editage.cn) for English language editing.

## Author Contributions

**Conceptualization:** Hongbin Gao.

**Data curation:** Hongbin Gao, Jingkuan Li.

**Formal analysis:** Hongbin Gao, Jingkuan Li.

**Funding acquisition:** Jingkuan Li.

**Methodology:** Hongbin Gao.

**Project administration:** Hongbin Gao, Jingkuan Li.

**Resources:** Hongbin Gao.

**Software:** Hongbin Gao.

**Supervision:** Hongbin Gao, Jingkuan Li.

**Visualization:** Hongbin Gao.

**Writing – original draft:** Hongbin Gao.

**Writing – review & editing:** Hongbin Gao.

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
