## [Decision Letter · Decision Letter 0]

19 Aug 2019

PONE-D-19-21511

Study on Co-Combustion of Coal Gangue and Polyvinyl Chloride

PLOS ONE

Dear Mr. Gao,

Thank you for submitting your manuscript to PLOS ONE. After careful consideration, we feel that it has merit but does not fully meet PLOS ONE’s publication criteria as it currently stands. Therefore, we invite you to submit a revised version of the manuscript that addresses the points raised during the review process.

ACADEMIC EDITOR: 

The novelty of the work must be further elaborated and highlighted in the paper.English must be improved. There are typos and grammatical errors in the text which need to be amended. What is the advantage of the PVC implemented in the present work in comparison with the other co-combusting agents?Provide state-of-the-art in the area by reviewing more relevant works. 

We would appreciate receiving your revised manuscript by Oct 03 2019 11:59PM. To enhance the reproducibility of your results, we recommend that if applicable you deposit your laboratory protocols in protocols.io, where a protocol can be assigned its own identifier (DOI) such that it can be cited independently in the future. For instructions see: http://journals.plos.org/plosone/s/submission-guidelines#loc-laboratory-protocols

We look forward to receiving your revised manuscript.

Kind regards,

Dr M. M. Sarafraz

PLOS ONE

Journal Requirements:

2. Please consider altering the title of the manuscript to reflect the main findings, and consider removing 'Study on'.

4.  We suggest you thoroughly copyedit your manuscript for language usage, spelling, and grammar. If you do not know anyone who can help you do this, you may wish to consider employing a professional scientific editing service.  

Reviewers' comments:

Reviewer's Responses to Questions

**Comments to the Author**

1. Is the manuscript technically sound, and do the data support the conclusions?

Reviewer #1: Yes

Reviewer #2: Partly

2. Has the statistical analysis been performed appropriately and rigorously? 

Reviewer #1: N/A

Reviewer #2: Yes

3. Have the authors made all data underlying the findings in their manuscript fully available?

Reviewer #1: Yes

Reviewer #2: No

4. Is the manuscript presented in an intelligible fashion and written in standard English?

Reviewer #1: No

Reviewer #2: Yes

5. Review Comments to the Author

Reviewer #1: 1- English must be reviewed and typos should be removed from the text. In its present form, there are grammatical errors, which need to be amended.

2- Introduction is superficial and authors are expected to reflect the novelty of the work, quality of their research. They have to show what has not been done in the literature and what would be the contribution of their work to the existing knowledge.

3- What is the novelty of the work? Any novelty in the methodology, mechanism of the experiment or modelling?

4- Abstract is full of abbreviation. It needs to be amended.

5- Part of this study is to conduct a series of experiments on co-combustion of coal and PVC. However, authors did not discuss the uncertainty and reliability of the devices, instruments and measurements in the paper. This is an experimental work and uncertainty analysis is heart of such a study. Discuss more about the accuracy of the sensors and systems.

6- Information in Table 2: why heating range is limited to 1000 C? What if temperature is ramped to 1500 C?

7- Any reactions between the crucible, moisture of the air with coal and PVC? How can it be identified?

8- Fig. 3a: What is the main reason for the fluctuation seen for heating rate 4C/min. This fluctuation is not seen for other heating rates.

9- Is Kissinger free model a suitable kinetic development model for this study?

Facing the above comments, my decision is major revision. Authors should highlight the novelty clearly in the text.

Reviewer #2: Commenting "Study on Co-Combustion of Coal Gangue and Polyvinyl Chloride"

1- English and writing style of the paper is weak. Authors must improve the quality of writing of this manuscript.

2- The aim and objective of this work is not clear. What authors are trying to address? What is the novelty of this work in comparison with the other papers?

3- The text is full of abbreviations and some of them have not been defined in the text. This must be corrected through out the manuscript.

4- Why there is a special focus on PVC as co-agent in the combustion of coal? What about other mixing agents? The advantage of PVC is not clear in the text.

5- Reliability calculations is required for the numbers and values reported in the paper. Moffat method could be one potential way to report the errors associated with the values reported in the paper.

6- Test rig or schematic diagram of the system used in the present research must be shown in the paper.

7- The methodology used in this work is not novel, thereby authors should justify the contribution of this work to the existing knowledge.

8- Conversion calculated in the present work must be compared to other combustion routes to evaluate the chemical performance of the co-combustion of coal and PVC.

9- Authors should discuss the level of oxygen available in the system. How far from the stoichiometric level?

10- CO2/CO ratio and ash analysis is required for the proposed system.

6. PLOS authors have the option to publish the peer review history of their article (what does this mean?). If published, this will include your full peer review and any attached files.

Reviewer #1: No

Reviewer #2: No

---

## [Author Response · Author response to Decision Letter 0]

3 Oct 2019

Responses to Reviewers’ Comments

We would like to thank the reviewers and editor for their help and efforts in providing constructive comments and valuable suggestions. Changes have been made to incorporate the suggestions of the academic editors and reviewers; these changes are marked in red in the revised manuscript. The changes and justifications for clarifying the reviewers’ comments are summarized as follows:

ACADEMIC EDITOR: 

1.The novelty of the work must be further elaborated and highlighted in the paper.

Response: Thank you for your suggestion. This study includes five novelties: (Ⅰ) The possibility of a treatment method based on co-combustion of coal gangue with polyvinyl chloride, which can be centrally recovered from municipal solid waste, is proposed. (Ⅱ) In order to analyze the interaction during the co-combustion of the two substances, a coupling analysis method for mixed combustion is presented, and the effectiveness of this method is verified by comparing with the correlation analysis results of co-combustion. (Ⅲ) Coal gangue and polyvinyl chloride exhibited very different combustion behaviors. (Ⅳ) The interactions during the co-combustion were observed. (Ⅴ) The co-combustion of the two substances is helpful to improve the problems of low calorific value and refractory burnout of coal gangue. The revised content is included in the “Abstract (on page 2),” “Introduction (on page 3),” and “Conclusions (on page 42).”

2.English must be improved. There are typos and grammatical errors in the text which need to be amended. 

Response: Thank you very much. According to your suggestion, this manuscript has been proofread by a professional English language editor (Editage). The revised contents have been marked in red.

3.What is the advantage of the PVC implemented in the present work in comparison with the other co-combusting agents? 

Response: Thank you for your suggestion. Increasing amounts of PVC are becoming available in municipal solid waste. The research team we worked with has been engaged in the treatment of solid waste for a long time and has recently studied the co-combustion characteristics of various municipal solid waste elements and coal gangue. As you have mentioned, our research shows that there are many co-combusting agents in municipal solid waste that have advantages when combusted along with coal gangue. However, in this paper, the reason for selecting PVC for co-combustion with coal gangue is that increasing amounts of PVC are becoming available in municipal solid waste from various industries, that is to say, PVC waste can be obtained easily for co-combustion with coal gangue.

4.Provide state-of-the-art in the area by reviewing more relevant works. 

Response: Thank you for your suggestion. The contributions and shortcomings of the previous research results related to this study and the work carried out in this study have been supplemented and improved. The revised content is provided in the “Introduction (on page 3)”.

Reviewer #1: 

1-English must be reviewed and typos should be removed from the text. In its present form, there are grammatical errors, which need to be amended.

Response: Thank you very much. According to your suggestion, this manuscript has been proofread by a professional English language editor. The revised contents have been marked in red.

2-Introduction is superficial and authors are expected to reflect the novelty of the work, quality of their research. They have to show what has not been done in the literature and what would be the contribution of their work to the existing knowledge.

Response: Thank you for your suggestion. The contributions and shortcomings of the previous research results related to this study and the work carried out in this study have been supplemented and improved. The revised content is provided in the “Introduction (on page 3)”.

3- What is the novelty of the work? Any novelty in the methodology, mechanism of the experiment or modelling?

Response: This study includes five novelties: (Ⅰ) The possibility of a treatment method based on co-combustion of coal gangue with polyvinyl chloride, which can be centrally recovered from municipal solid waste, is proposed. (Ⅱ) In order to analyze the interaction during the co-combustion of the two substances, a coupling analysis method for mixed combustion is presented, and the effectiveness of this method is verified by comparing with the correlation analysis results of co-combustion. (Ⅲ) Coal gangue and polyvinyl chloride exhibited very different combustion behaviors. (Ⅳ) The interactions during the co-combustion were observed. (Ⅴ) The co-combustion of the two substances is helpful to improve the problems of low calorific value and refractory burnout of coal gangue. Thus, the main novelties are the object of analysis and the analytical method.

4- Abstract is full of abbreviation. It needs to be amended.

Response: Thank you for your suggestion. The content of the abstract has been refined and improved, and all the abbreviations have been expanded. 

5- Part of this study is to conduct a series of experiments on co-combustion of coal and PVC. However, authors did not discuss the uncertainty and reliability of the devices, instruments and measurements in the paper. This is an experimental work and uncertainty analysis is heart of such a study. Discuss more about the accuracy of the sensors and systems.

Response: Thank you for your suggestion. Content related to the accuracy of the experimental equipment have been supplemented. At the same time, in order to improve the accuracy of the experimental results, the experimental system was calibrated before starting the experiment, and the relevant contents have been added to the paper. The revised content is present in the section “Selection of experimental equipment and system design” (on page 7). 

6- Information in Table 2: why heating range is limited to 1000℃? What if temperature is ramped to 1500℃?

Response: Thank you for your suggestion. The burnout temperature of coal gangue and plastics is 500 ℃ – 600 ℃. The final results of the experiments in this paper also show that the burnout temperature of both mixed combustion and individual combustion is lower than 600 ℃, so the heating final temperature of 1000 ℃ during the test can completely meet the needs. If the temperature increases to 1500 ℃, there is no difference between the law of temperature parameter change and the conclusion of mixed combustion effect obtained in this paper. This explanation is also added in the article. The revised content is present in the section “Single-component combustion experiment” (on page 9).

7- Any reactions between the crucible, moisture of the air with coal and PVC? How can it be identified?

Response: Thank you for your suggestion. In order to determine whether any chemical reaction of coal gangue with PVC and moisture occurred during the mixed combustion, the composition of flue gas and ash slag was specifically analyzed. The results show that there is no new chemical composition. This part of the research content is not included in this paper; it will be specifically analyzed in other subsequent articles. Three pictures of the results of the experiments are shown below.

CO2 and H2O emission from CG combustion 

CO2 and H2O emission from PVC combustion

CO2 and H2O emission from CG and PVC co-combustion

8- Fig. 3a: What is the main reason for the fluctuation seen for heating rate 4C/min. This fluctuation is not seen for other heating rates.

Response: Thank you for your suggestion. According to our analysis, the distribution of reactive materials in the crucible is uneven, the thickness is different, the reaction time is longer, and the reaction rate is slow when the heating rate is low, which reflects this objective influence. Due to the acceleration of heating rate and the short reaction time, this effect is not easy to be observed and shows a smooth curve. Because of the small fluctuation on the global curve, it is similar to the clutter in the signal source, irrespective of whether smoothing is applied, but this has no effect on the experimental trend and conclusion. Thank you very much for this question. We had not considered this fluctuation before; we will pay attention to this aspect in the future.

9- Is Kissinger free model a suitable kinetic development model for this study?

Facing the above comments, my decision is major revision. Authors should highlight the novelty clearly in the text.

Response: Thank you for your suggestion. Yes, Kissinger free model is a suitable kinetic development model for this study. 

The distributed activation energy model used in this study can be expressed as follows:

Kissinger free model can be expressed as follows:

It can be seen that the linear cluster obtained by the distributed activation energy model is 0.6075 times larger than that obtained by Kissinger free model, and the final results obtained by these two methods are very close.

Reviewer #2:

1- English and writing style of the paper is weak. Authors must improve the quality of writing of this manuscript.

Response: Thank you very much. According to your suggestion, the abstract and text have been modified and perfected, and the spelling and grammar of the entire text have been modified and reviewed by a professional English language editor. The revised contents have been marked in red.

2- The aim and objective of this work is not clear. What authors are trying to address? What is the novelty of this work in comparison with the other papers?

Response: Thank you for your suggestion. The aim and objective of this work is to analyze the influence of the ratio of PVC in the mixture on the temperature parameters, activation energy, and interaction of co-combustion. This paper aims to find out the co-combustion effect of CG and PVC. This study includes five novelties: (Ⅰ) The possibility of a treatment method based on the co-combustion of coal gangue with polyvinyl chloride, which can be centrally recovered from municipal solid waste, is proposed. (Ⅱ) In order to analyze the interaction during the co-combustion of the two substances, a coupling analysis method for mixed combustion is presented, and the effectiveness of this method is verified by comparing with the correlation analysis results of co-combustion. (Ⅲ) Coal gangue and polyvinyl chloride exhibited very different combustion behaviors. (Ⅳ) The interactions during the co-combustion were observed. (Ⅴ) The co-combustion of the two substances is helpful to improve the problems of low calorific value and refractory burnout of coal gangue. The revised content is provided in the “Introduction (on page 3)”.

3- The text is full of abbreviations and some of them have not been defined in the text. This must be corrected through out the manuscript.

Response: Thank you for your suggestion. The text has been modified and perfected, and all these problems have been dealt with. The abbreviations have been defined in “abbreviations (on page 43)”.

4- Why there is a special focus on PVC as co-agent in the combustion of coal? What about other mixing agents? The advantage of PVC is not clear in the text.

Response: Thank you for your suggestion. Increasing amounts of PVC are becoming available in municipal solid waste. The research team we worked with has been engaged in the treatment of solid waste for a long time and has recently studied the co-combustion characteristics of various municipal solid waste elements and coal gangue. As you have mentioned, our research shows that there are many co-combusting agents in municipal solid waste that have advantages when combusted along with coal gangue. However, in this paper, the reason for selecting PVC for co-combusting with coal gangue is that increasing amounts of PVC are becoming available in municipal solid waste from various industries, that is to say, PVC waste can be obtained easily for co-combustion with coal gangue.

5- Reliability calculations is required for the numbers and values reported in the paper. Moffat method could be one potential way to report the errors associated with the values reported in the paper.

Response: Thank you for your suggestion. We are sorry that we have not found any relevant information about Moffat method. We hope that you can provide us with some relevant literature or reference about Moffat method.

6- Test rig or schematic diagram of the system used in the present research must be shown in the paper.

Response: Thank you for your suggestion. The schematic diagram and structure diagram of the test system have been added to the paper, as shown in Figs. 1(a) and (b), respectively.

7- The methodology used in this work is not novel, thereby authors should justify the contribution of this work to the existing knowledge.

Response: Thank you for your suggestion. The purpose of this paper is to study the co-combustion effect of CG and PVC. In order to achieve this goal, the TG experimental curves of three proportions of CG and PVC blends were evaluated and compared with those of CG and PVC. The influence of the PVC ratio in the mixture on the temperature parameters, activation energy, and interaction of co-combustion were analyzed. In order to effectively analyze the interaction between CG and PVC in the overall co-combustion process, an analysis method for the coupling of the two combustion processes, which is based on the fact that the variance between the actual value and the mean value can be calculated by the least-squares method, is presented. The analysis results are obtained, and the co-combustion effect of CG and PVC is evaluated according to the analysis results. 

8- Conversion calculated in the present work must be compared to other combustion routes to evaluate the chemical performance of the co-combustion of coal and PVC.

Response: Thank you for your suggestion. Our focus is on the influence of the PVC ratio in the mixture on the temperature parameters, activation energy, and interaction of co-combustion. The research ideas you have put forward provide a direction for us to carry out future research. The revised content is provided in the “Introduction (on page 3)”.

9- Authors should discuss the level of oxygen available in the system. How far from the stoichiometric level?

Response: Thank you for your suggestion. In this study, about 18 mg oxygen is needed for 10 mg gangue combustion and about 32 mg oxygen is needed for 10 mg PVC combustion. Taking a heating rate of 30 ℃/min and an air flow rate of 60 mL/min as an example, when the temperature rises from 50 ℃ to 1000 ℃, the system requires 300 mg oxygen. It can be seen that the experimental gas supply is much larger than the amount of air needed for combustion. This content has been added to the section “Mixed-component combustion experiment (on page 10).”

10- CO2/CO ratio and ash analysis is required for the proposed system.

Response: Thank you for your suggestion. The characteristics of flue gas and ash in the mixed combustion are also analyzed. Here are some pictures of the results of the related experiments. This part of the research content is not included in this paper; it will be specifically analyzed in other subsequent articles. 

CO2 and H2O emission from CG combustion 

CO2 and H2O emission from PVC combustion

CO2 and H2O emission from CG and PVC co-combustion

Infrared spectra of ash after combustion of CG, PVC, and mixtures

---

## [Decision Letter · Decision Letter 1]

14 Oct 2019

Thermogravimetric analysis of the co-combustion of coal and polyvinyl chloride

PONE-D-19-21511R1

Dear Dr. Gao,

We are pleased to inform you that your manuscript has been judged scientifically suitable for publication and will be formally accepted for publication once it complies with all outstanding technical requirements.

With kind regards,

Mason Sarafraz

Academic Editor

PLOS ONE

Additional Editor Comments (optional):

Reviewers' comments:

Reviewer's Responses to Questions

**Comments to the Author**

1. If the authors have adequately addressed your comments raised in a previous round of review and you feel that this manuscript is now acceptable for publication, you may indicate that here to bypass the “Comments to the Author” section, enter your conflict of interest statement in the “Confidential to Editor” section, and submit your "Accept" recommendation.

Reviewer #1: All comments have been addressed

Reviewer #2: All comments have been addressed

2. Is the manuscript technically sound, and do the data support the conclusions?

Reviewer #1: Yes

Reviewer #2: Yes

3. Has the statistical analysis been performed appropriately and rigorously? 

Reviewer #1: Yes

Reviewer #2: Yes

4. Have the authors made all data underlying the findings in their manuscript fully available?

Reviewer #1: Yes

Reviewer #2: Yes

5. Is the manuscript presented in an intelligible fashion and written in standard English?

Reviewer #1: Yes

Reviewer #2: Yes

6. Review Comments to the Author

7. PLOS authors have the option to publish the peer review history of their article (what does this mean?). If published, this will include your full peer review and any attached files.

Reviewer #1: No

Reviewer #2: No

---

## [Editor Report · Acceptance letter]

18 Oct 2019

PONE-D-19-21511R1 

Thermogravimetric analysis of the co-combustion of coal and polyvinyl chloride 

Dear Dr. Gao:

I am pleased to inform you that your manuscript has been deemed suitable for publication in PLOS ONE. Congratulations! Your manuscript is now with our production department. 

With kind regards,

on behalf of

Dr. Mason Sarafraz 

Academic Editor

PLOS ONE